# A covalent PIN1 inhibitor selectively targets cancer cells by a dual mechanism of action

Elena Campaner[1,2,*], Alessandra Rustighi[1,*], Alessandro Zannini[1,2], Alberto Cristiani[1], Silvano Piazza[1,†], Yari Ciani[1], Ori Kalid[3,†], Gali Golan[3,†], Erkan Baloglu[3], Sharon Shacham[3], Barbara Valsasina[4], Ulisse Cucchi[4], Agnese Chiara Pippione[5], Marco Lucio Lolli[5], Barbara Giabbai[6], Paola Storici[6], Paolo Carloni[7], Giulia Rossetti[7,8,9], Federica Benvenuti[10], Ezia Bello[11], Maurizio D'Incalci[11], Elisa Cappuzzello[12], Antonio Rosato[12,13] & Giannino Del Sal[1,2]

The prolyl isomerase PIN1, a critical modifier of multiple signalling pathways, is overexpressed in the majority of cancers and its activity strongly contributes to tumour initiation and progression. Inactivation of PIN1 function conversely curbs tumour growth and cancer stem cell expansion, restores chemosensitivity and blocks metastatic spread, thus providing the rationale for a therapeutic strategy based on PIN1 inhibition. Notwithstanding, potent PIN1 inhibitors are still missing from the arsenal of anti-cancer drugs. By a mechanism-based screening, we have identified a novel covalent PIN1 inhibitor, KPT-6566, able to selectively inhibit PIN1 and target it for degradation. We demonstrate that KPT-6566 covalently binds to the catalytic site of PIN1. This interaction results in the release of a quinone-mimicking drug that generates reactive oxygen species and DNA damage, inducing cell death specifically in cancer cells. Accordingly, KPT-6566 treatment impairs PIN1-dependent cancer phenotypes in vitro and growth of lung metastasis in vivo.

[1] National Laboratory CIB (LNCIB), Area Science Park Padriciano, Trieste 34149, Italy. [2] Department of Life Sciences, University of Trieste, Trieste 34127, Italy. [3] Karyopharm Therapeutics, Newton, Massachusetts 02459, USA. [4] Nerviano Medical Sciences Srl, Nerviano 20014, Italy. [5] Department of Science and Drug Technology, University of Torino, Torino 10125, Italy. [6] Elettra Sincrotrone Trieste S.C.p.A., Area Science Park Basovizza, Trieste 34149, Italy. [7] Computational Biomedicine, Institute for Advanced Simulation (IAS-5) and Institute of Neuroscience and Medicine (INM-9), Forschungszentrum Jülich, Jülich 52425, Germany. [8] Jülich Supercomputing Center (JSC), Forschungszentrum Jülich, Jülich 52425, Germany. [9] Department of Oncology, Hematology and Stem Cell Transplantation, University Hospital Aachen, RWTH Aachen University, Aachen 52074, Germany. [10] International Centre for Genetic Engineering and Biotechnology (ICGEB), Area Science Park Padriciano, Trieste 34149, Italy. [11] IRCCS-Mario Negri Institute for Pharmacological Research, Milano 20156, Italy. [12] Department of Surgery, Oncology and Gastroenterology, Oncology and Immunology Section, University of Padova, Padova 35128, Italy. [13] Veneto Institute of Oncology (IOV)-IRCCS, Padova 35128, Italy. * These authors contributed equally to this work. † Present addresses: Bioinformatics Core Facility, Center for Integrative Biology (CIBIO), University of Trento, Trento 38123, Italy (S.P.); Pi Therapeutics, P.O.B. 4044, Ness Ziona 7403635, Israel (O.K.); Evogene Ltd., P.O.B. 2100, Rehovot 7612002, Israel (G.G.). Correspondence and requests for materials should be addressed to G.D.S. (email: delsal@lncib.it).

Phosphorylation of proteins at serine or threonine residues followed by proline (S/T-P) represents a common and central signal transduction mechanism in many oncogenic pathways and it is executed by a repertoire of proline-directed kinases, for example, Cyclin-dependent kinases, and Mitogen activated protein kinases, that fulfil key roles in controlling signal transduction. Numerous oncogenes and tumour suppressors either are directly regulated by and/or trigger signalling pathways involving such phosphorylation events[1].

In proteins, S/T-P motifs can adopt either a *cis* or a *trans* conformation. Spontaneous conversion between isomers occurs at a very slow rate and is further slowed down by phosphorylation of these motifs. However, phospho-S/T-P sites can be recognized by the peptidyl-prolyl *cis/trans* isomerase (PPIase) PIN1, which catalyses *cis-trans* or *trans-cis* conformational changes around the S-P or T-P bond. Among PPIases, PIN1 is the only enzyme able to efficiently bind proteins containing phosphorylated S/T-P sites[1]. Targeting of these motifs occurs in a modular fashion: PIN1 firstly binds them through its WW domain, and then catalyses their *cis/trans* isomerization through its catalytic PPIase domain. Importantly, as a consequence of their modified shape, PIN1 client proteins are profoundly affected in terms of stability, subcellular localization, interaction with cellular partners and occurrence of other post-translational modifications on them[2]. Notably, PIN1 controls the ability of many transcription factors to interact with their partners on gene promoters and instructs transcription complexes towards specific gene expression profiles[3].

PIN1 has been shown to play a critical role during oncogenesis[4]. It is overexpressed in the majority of cancers and acts as a modulator of several cancer-driving signalling pathways, including c-MYC, NOTCH1, WNT/β-catenin and RAS/MEK/ERK pathways, while it simultaneously curbs several tumour suppressors[5]. Work done by us has shown that PIN1 enables a mutant p53 (mut-p53) pro-metastatic transcriptional program and boosts breast cancer stem cells (CSCs) expansion through activation of the NOTCH pathway[6,7].

Genetic ablation of PIN1 reduces tumour growth and metastasis in several oncogene-induced mouse models of tumorigenesis, indicating the requirement for PIN1 for the development and progression of some tumours[4]. In addition, PIN1 inhibition sensitizes breast cancer cells to different targeted- and chemo-therapies[8–10] or overcomes drug resistance[7,11]. Accordingly, PIN1 inhibition alone has been recently shown to curb both leukaemia and breast cancer stem cells by simultaneously dampening multiple oncogenic pathways[7,12,13]. Altogether these data strongly indicate that targeting PIN1 dismantles oncogenic pathway cooperation in CSCs and non-CSC tumour cells, providing a rationale for the development of PIN1 targeted therapies. A number of features, including its well-defined active site, its high specificity and its low expression in normal tissues, make PIN1 an attractive target for the design of small molecule inhibitors[5,14]. However, its small and shallow enzymatic pocket, as well as the requirement of a molecule with a negatively charged moiety for interfacing with its catalytic centre have been challenging the design of PIN1 inhibitors[14]. Although many molecules, mainly non-covalent inhibitors, have been isolated so far, none of them has reached the clinical trial phase because of their unsatisfactory pharmacological performance in terms of potency, selectivity, solubility, cell permeability and stability[5,14].

In this work we describe a novel PIN1 inhibitor identified from a library of commercial compounds we screened to isolate PIN1 inhibitors with increased biochemical efficiency based on a covalent mechanisms of action[15]. The compound 2-{[4-(4-*tert*-butylbenzene-sulfonamido)-1-oxo-1,4-dihydronaphthalen-2-yl]sulfanyl}acetic acid, hereafter called KPT-6566 (1), turned out to selectively inhibit *in vitro*

the catalytic activity of PIN1. Structural, biochemical and cell-based experiments allowed us to establish the mechanism of action of this compound which, acting both as a covalent PIN1 inhibitor and as a PIN1-activated cytotoxic agent, is able to specifically kill PIN1-proficient tumour cells while leaving normal cells unaffected.

## Results

**Structure- and mechanism-based screening for PIN1 inhibitors.** With the intent of isolating covalent inhibitors targeting the cysteine C113 residue of PIN1 catalytic core, we screened a drug like collection of 200,000 commercial compounds obtained from several drug repositories (Fig. 1a). The compound pool was first filtered applying the Lipinski's rule of five criteria for enhanced drug-likeness. Then, a virtual structure-based screening was performed using the crystal structure of human PIN1 (PDB entry 2XPB)[16]. The compounds showing the higher docking scores were then subjected to another virtual screening specifically designed to identify compounds able to covalently target the active site residue C113. To this aim, a covalent docking approach using the CovDock-VS method[17] was exploited. These approaches yielded around one hundred possible PIN1 covalent binders that were tested afterwards for cytotoxicity against melanoma A375 cells using the MTT viability assay. Non-transformed 3T3 cells were used as a control to make sure hit compounds were not generally cytotoxic. Nine compounds were selected for their differential toxicity between A375 and 3T3 cells (Supplementary Table 1) and were chosen for further characterization as potential PIN1 inhibitors.

**KPT-6566 inhibits PIN1 PPIase activity through covalent binding.** The compounds derived from the above screening were initially tested for their capacity to inhibit the catalytic activity of recombinant human PIN1 in a trypsin-coupled peptidyl-prolyl isomerization assay (PPIase assay)[18,19]. The mutant PIN1 S67E protein was used as control of a reduced PPIase activity[20]. The results show that, out of nine tested compounds, only KPT-6566 (1) inhibited the PPIase activity of PIN1 (Fig. 1b) and turned out to have an IC50 of 0.64 μM (Supplementary Table 2). From a chemical point of view, KPT-6566 contains a polycyclic aromatic hydrocarbon substituted with a sulfanyl-acetic acid and a *tert*-butylphenylsulfonamide moiety (Fig. 1c), and it was synthesized for subsequent analyses (Supplementary Methods).

To demonstrate whether KPT-6566 covalently binds to PIN1, recombinant human PIN1 protein was analysed by mass spectrometry following incubation with KPT-6566 or DMSO. Liquid chromatography/mass spectrometry (LC/MS) analysis revealed a 90 Da increase of PIN1 molecular weight (MW) upon treatment with KPT-6566 (Fig. 1d), indicating that a modification by addition of a sulfanyl-acetate group (-$S$-$CH_2$-COOH) had occurred. The $+90$ Da adduct was not detected when C113 was mutated to alanine (C113A, Supplementary Fig. 1a), indicating that C113 was the target of this covalent modification.

To confirm the above results, PIN1 was incubated with KPT-6566 or DMSO, and subjected to trypsin digestion. The originated peptides were then analysed by Matrix-assisted laser desorption ionization–time of flight (MALDI-ToF-ToF) MS and MS/MS. MS spectrum showed the presence of a peptide at 2213.9 m z$^{-1}$, corresponding to the S98–K117 peptide of PIN1 bearing a $+90$ Da MW increase (Supplementary Fig. 1b). MS/MS fragmentation of the peptide at 2213.9 m z$^{-1}$ performed with MALDI-ToF-ToF unambiguously confirmed the identity of the peptide and the presence of a 90 Da adduct on C113 (Supplementary Fig. 1c). These MS analyses together indicate the covalent addition of the 90 Da sulfanyl-acetate group of

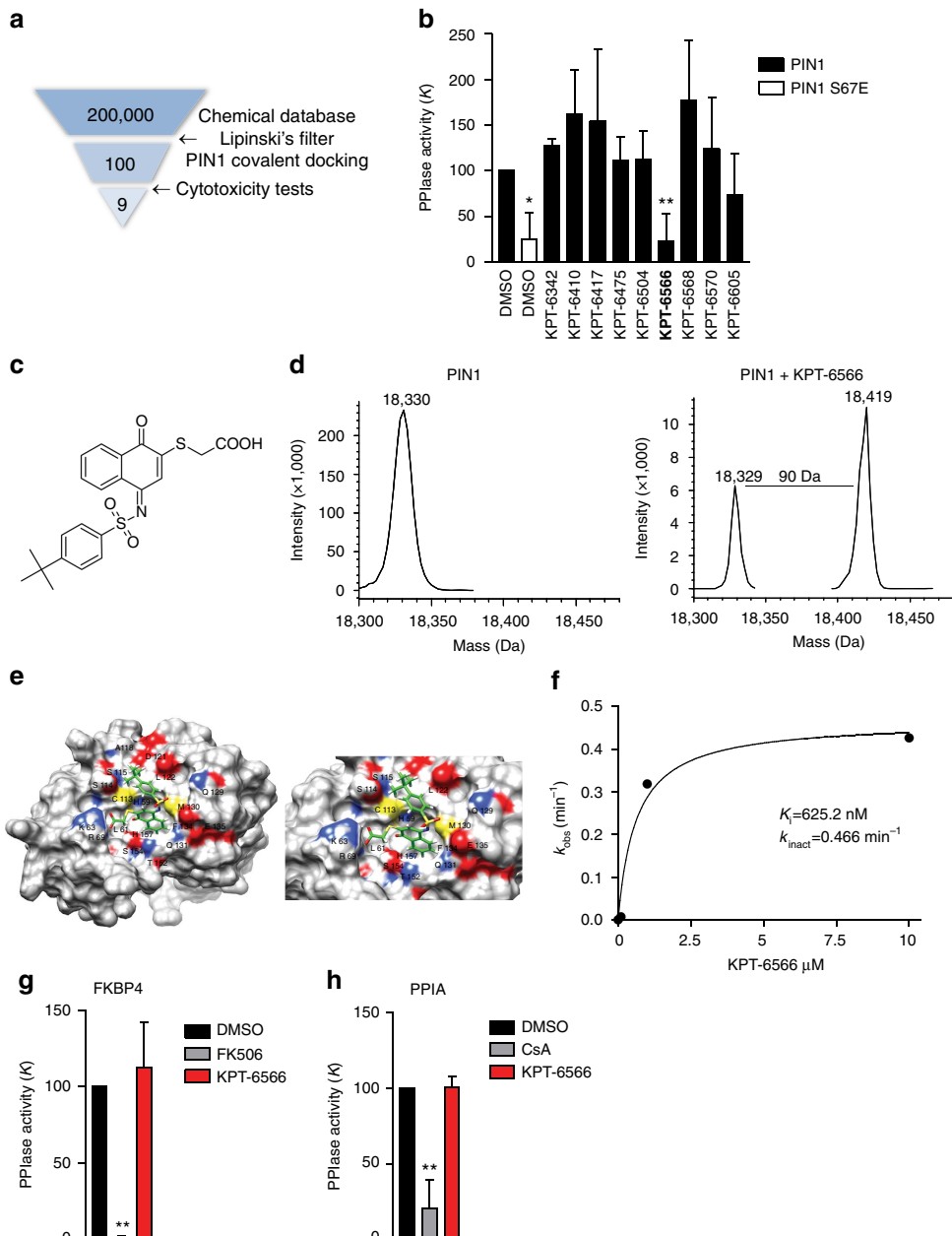

**Figure 1 | KPT-6566 specifically inactivates PIN1 PPIase activity *in vitro*.** (**a**) Scheme representing the screening conducted with the drug like collection, indicating the stepwise approach that ends up with nine potential covalent inhibitors of PIN1. (**b**) Bar plot indicating PPIase activity $K$ of PIN1 (black), PIN1 S67E (white) and PIN1 with 30 μM of the indicated compounds. Positive hit is marked in bold. (**c**) Chemical structure of KPT-6566 (1): the compound contains a polycyclic aromatic hydrocarbon bound to a sulfanyl-acetic acid group (-$S$-$CH_2$-COOH) and to a *tert*-butylphenyl group through a sulfonamide moiety ($N$-$SO_2$). (**d**) Mass spectrum deconvolution of DMSO (left) and KPT-6566 treated (right) PIN1. A MW increase of 90 Da can be appreciated in the compound treated sample. (**e**) 3D image showing the docking of KPT-6566 in the catalytic pocket of PIN1. KPT-6566 is shown in green licorice representation. The surface of PIN1 is shown as solid representation and coloured by heteroatom. The binding cavity is magnified in the right insert. (**f**) Plot of the observed rate constants for inhibition ($k_{obs}$) against inhibitor concentration of KPT-6566 from which estimations of kinetic parameters for covalent inhibition of PIN1 *in vitro* were made. The corresponding $k_{inact}$ and $K_i$ values are reported. (**g**) Bar plot indicating PPIase activity of GST-FKBP4 incubated with DMSO, 30 μM FK506 or 30 μM KPT-6566. (**h**) Same as in (**g**) for GST-PPIA incubated with DMSO, 30 μM Cyclosporin A or 30 μM KPT-6566. Data shown in **b,g,h**, are the means ± s.d. of $n = 3$ independent experiments, *$P < 0.05$, **$P < 0.01$; two-tailed Student's *t*-test.

KPT-6566 to the sulfur atom of C113, through a disulfide bond. Accordingly, when the peptide-KPT-6566 mixture was incubated with 100 μM of the reducing agent dithiothreitol (DTT), the 90 Da adduct was not detected on the 2213.9 m z⁻¹ peptide, confirming that the sulfur atom of the sulfanyl-acetate moiety is involved in a disulfide bridge with the -SH group of C113 (Supplementary Fig. 1b).

Next, to assess the mode of binding of KPT-6566 to PIN1, we performed a molecular docking using Glide from the Schrödinger suite[21–23] (Supplementary Table 3). In accordance with MS results, the best docking pose (Pose 1) obtained within this analysis suggests that KPT-6566 positions itself into the catalytic pocket of human PIN1 by creating (i) a hydrogen bond with the side chains of K63 and R69, the active site residues mediating

phosphate binding, and (ii) hydrophobic contacts with some residues involved in the recognition of the proline substrate[24]. Importantly, the electrophile sulfanyl-acetate moiety of the compound directly faces the nucleophile sulfur atom of C113 (Fig. 1e and Supplementary Fig. 1d), which might lead to a covalent complex formation as observed in the MS analyses.

Considering the covalent mechanism of action of KPT-6566, we measured its potency as $k_{inact}/K_i$ ratio, where $K_i$ describes the affinity of the initial non-covalent interaction and $k_{inact}$ is the rate of the subsequent bond-forming reaction[15]. The catalytic activity of PIN1 was measured in the presence of increasing KPT-6566 concentrations at several time points after preincubation (Fig. 1f and Supplementary Fig. 1e). Compared to other covalent PIN1 inhibitors such as Juglone[25], KPT-6566 shows a higher potency ($k_{inact}/K_i = 745.4\,min^{-1}\,nM^{-1}$) due to a high $k_{inact}$ ($0.466 \pm 0.05781\,min^{-1}$), despite its affinity is in a high nanomolar range ($K_i = 625.2 \pm 324.7\,nM$) (Supplementary Table 4).

Finally, to determine whether KPT-6566 selectively inhibits PIN1 and not other PPIases, we measured the impact of KPT-6566 on the PPIase activity of recombinant GST-FKBP4 and GST-PPIA, two isomerases also containing cysteine residues and belonging to the FKBP and Cyclophilin families of PPIases[26] (Supplementary Fig. 1f). Neither of these PPIases was affected by KPT-6566, while, their specific inhibitors, FK506 and Cyclosporin A (CsA) respectively, abolished their catalytic activity (Fig. 1g,h).

**KPT-6566 impacts on PIN1 cellular functions.** To assess whether KPT-6566 could inhibit PIN1 function, we treated immortalized fibroblasts derived from wild-type (WT, $Pin1^{+/+}$) or $Pin1$ knockout (KO, $Pin1^{-/-}$) mouse embryos (MEFs)[6,27] with increasing amounts of KPT-6566, and monitored cell proliferation. In WT MEFs KPT-6566 had a negative, dose-dependent effect on proliferation (Fig. 2a) and induced a decrease of hyperphosphorylated pRB and Cyclin D1 levels (Fig. 2b). In $Pin1$ KO MEFs, instead, KPT-6566 treatment had no effect on proliferation even at the highest dose, and the levels of both Cyclin D1 and hyperphosphorylated pRB were unaffected (Fig. 2a,b). Ectopic expression of HA-tagged PIN1 re-sensitized $Pin1$ KO MEFs to KPT-6566 and, accordingly, Cyclin D1 levels were decreased (Fig. 2c,d). In empty-vector expressing $Pin1$ KO MEFs, instead, KPT-6566 had only a slight and statistically not significant effect on proliferation and no impact on Cyclin D1.

We next tested the effects of increasing doses of KPT-6566 on an isogenic cell model constituted of (i) non-transformed MCF10A mammary epithelial cells, which express lower levels of PIN1 than cancer cell lines, and in which the enzyme is mainly inactivated due to S71 phosphorylation[12], and (ii) H-RasV12-transformed MCF10AT1 cells[28] which, instead, express high levels of active PIN1 (Fig. 2e, left). Viability assays highlighted that, consistent with the levels of PIN1, KPT-6566 was four times more potent towards MCF10AT1 than towards MCF10A cells (Fig. 2e, right and Supplementary Table 5).

To test whether sensitivity to KPT-6566 represents a common trait of cancer cells with respect to normal cells, we treated cancer cell lines of different origin and normal breast epithelial cells with increasing doses of the drug and measured their viability. All tested cancer cells expressed higher levels of PIN1 (Fig. 2f) and were more sensitive to KPT-6566, even at low micromolar concentrations (Fig. 2g).

**KPT-6566 impacts on levels and function of PIN1 substrates.** PIN1 influences the turnover and activity of various proteins, including Cyclin D1, p65 (NF-κB), c-JUN, NOTCH1 intracellular domain (N1-ICD) and MCL-1 (ref. 2). To understand the impact of KPT-6566 as a PIN1 inhibitor, we explored the effect of its administration on these PIN1 targets in MDA-MB-231 cells. The effect of KPT-6566 treatment on the levels of these PIN1 client proteins was comparable to that observed following $PIN1$ silencing (Fig. 3a and Supplementary Fig. 2a,b). We also confirmed downregulation of MCL-1 upon KPT-6566 treatment in lung, prostate and pancreatic cancer cell lines (Supplementary Fig. 2c).

To confirm the above observations at a functional level, we tested the effect of KPT-6566 on the activation of two pathways that are controlled by PIN1, the mut-p53 and NOTCH1 pathways[6,7]. We evaluated the effect of KPT-6566 on the transcription of a selection of mut-p53 and NOTCH1 target genes. RT-qPCR analyses of MDA-MB-231 cells treated with increasing concentrations of KPT-6566 showed a dose-dependent decrease in their expression levels (Fig. 3b). This effect was comparable to that observed upon $PIN1$ silencing (Supplementary Fig. 2d). Altogether these data showed that, by inactivating PIN1, KPT-6566 elicits the downregulation of PIN1 substrates and of their target genes.

**KPT-6566 impairs PIN1-dependent oncogenic phenotypes.** Next, we tested whether KPT-6566 interferes with colony forming efficiency of transformed cells[29,30]. We first assessed the $IC_{50}$ of KPT-6566 by dose escalation studies in MDA-MB-231 cells (Supplementary Fig. 3a). The $IC_{50}$ for these experiments turned out to be 1.2 μM. We used this concentration to treat MDA-MB-231 cells transfected with control- or $PIN1$ siRNA (Fig. 3c). Upon KPT-6566 treatment, we observed an inhibition of colony formation in MDA-MB-231 cells transfected with control siRNA. Similar inhibitory effects were obtained with another PIN1 inhibitor, PiB[31]. Instead in $PIN1$ silenced cells, which per se show a decreased colony forming efficiency, KPT-6566 did not further affect colony number, while PiB caused additional impairment of the colony forming efficiency, indicating off-target effects of this compound[31] (Fig. 3c). The specific effect of KPT-6566 was confirmed by using another siRNA sequence (Supplementary Fig. 3b,c). Superimposable results were also obtained with the prostate cancer cell line PC3 (Supplementary Fig. 3d–f). Altogether these data suggest that KPT-6566 impairs colony formation through specific PIN1 inhibition.

Treatment of MDA-MB-231 and PC3 cancer cells with KPT-6566 strongly impaired also the migratory and invasive properties of these cells, while having only moderate effects on proliferation in the short time-frame of the experiment (Fig. 3d and Supplementary Fig. 3g).

**KPT-6566 curbs self-renewal of breast cancer stem cells.** We and others have recently demonstrated that silencing or inhibition of PIN1 impairs the maintenance of mammary CSCs of both human and mouse origin[7,13]. To evaluate the impact of KPT-6566 administration on breast CSCs maintenance, secondary mammosphere (M2) formation assays were performed in MDA-MB-231 cells. Cells were treated with KPT-6566 or DMSO, prepared as single-cell suspensions and grown in low-attachment conditions to form M2, as previously done[7]. KPT-6566 treatment showed a significant inhibitory effect on M2 formation compared to DMSO (Fig. 3e and Supplementary Fig. 3h). This result was further confirmed by the decrease of the CD44$^+$/CD24$^{-/low}$/ESA$^+$ breast CSC enriched population[32] and reduction of the levels of three different stem cell markers[33,34] following KPT-6566 treatment (Supplementary Fig. 3i and Fig. 3f). To evaluate the requirement of PIN1 for these effects, we tested the M2 formation efficiency (M2FE) of

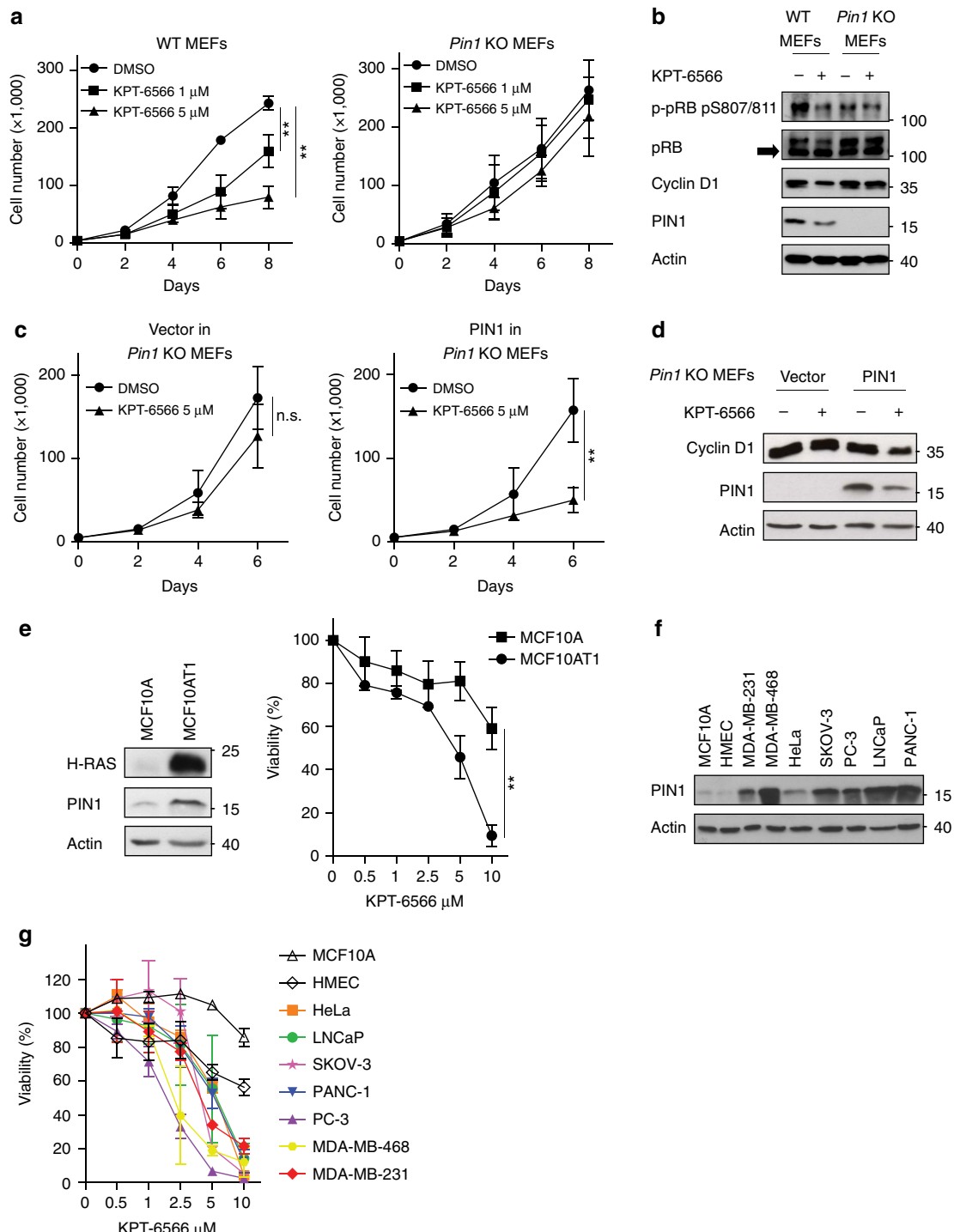

**Figure 2 | KPT-6566 impacts on cell proliferation and viability in a PIN1-dependent manner.** (**a**) Growth curves of WT (left) or *Pin1* KO (right) MEFs treated with the indicated concentrations of KPT-6566 or DMSO. (**b**) Immunoblotting of the indicated cell cycle-related proteins in WT or *Pin1* KO MEFs treated with 5 μM KPT-6566 ( + ) or DMSO ( − ) for 48 h. (**c**) Growth curves in empty-vector transduced *Pin1* KO MEFs (Vector, left) or *Pin1* KO MEFs reconstituted with HA-PIN1 (PIN1, right) treated with KPT-6566 or DMSO. (**d**) Immunoblotting of the indicated proteins in *Pin1* KO MEFs or *Pin1* KO MEFs reconstituted with HA-PIN1 treated with 5 μM KPT-6566 ( + ) or DMSO ( − ) for 48 h. (**e**) Left, immunoblotting of the indicated proteins in cell lysates from MCF10A and MCF10AT1 cells. Right, cell viability (WST) assay of MCF10A and MCF10AT1 treated with the indicated concentrations of KPT-6566 for 48 h. (**f**) Immunoblotting of PIN1 in normal breast epithelial cells (MCF10A, HMEC) and in the indicated cancer cell lines. (**g**) Cell viability (ATPlite) assay of the same cell lines as in (**f**) treated with the indicated concentrations of KPT-6566 for 48 h. **b,d**–**f** Actin levels are reported as loading control; size markers are indicated. Data shown in **a,c,e,g** are the means ± s.d. of $n = 3$ independent experiments, **$P < 0.01$, n.s. not significant; two-tailed Student's $t$-test.

MCF10AT1 cells transduced with control- or *PIN1* short hairpin RNA (shRNA) expressing vectors, and treated with KPT-6566, PiB, or DMSO. sh*PIN1* expressing cells showed a twofold decrease in M2FE compared to control shRNA expressing cells (Fig. 3g and Supplementary Fig. 3j,k). Treatment of control cells with PiB or KPT-6566 caused a similar decrease of M2FE. In contrast to PiB, KPT-6566 had no impact in *PIN1* silenced cells, further supporting a specific PIN1 inhibitory activity of this

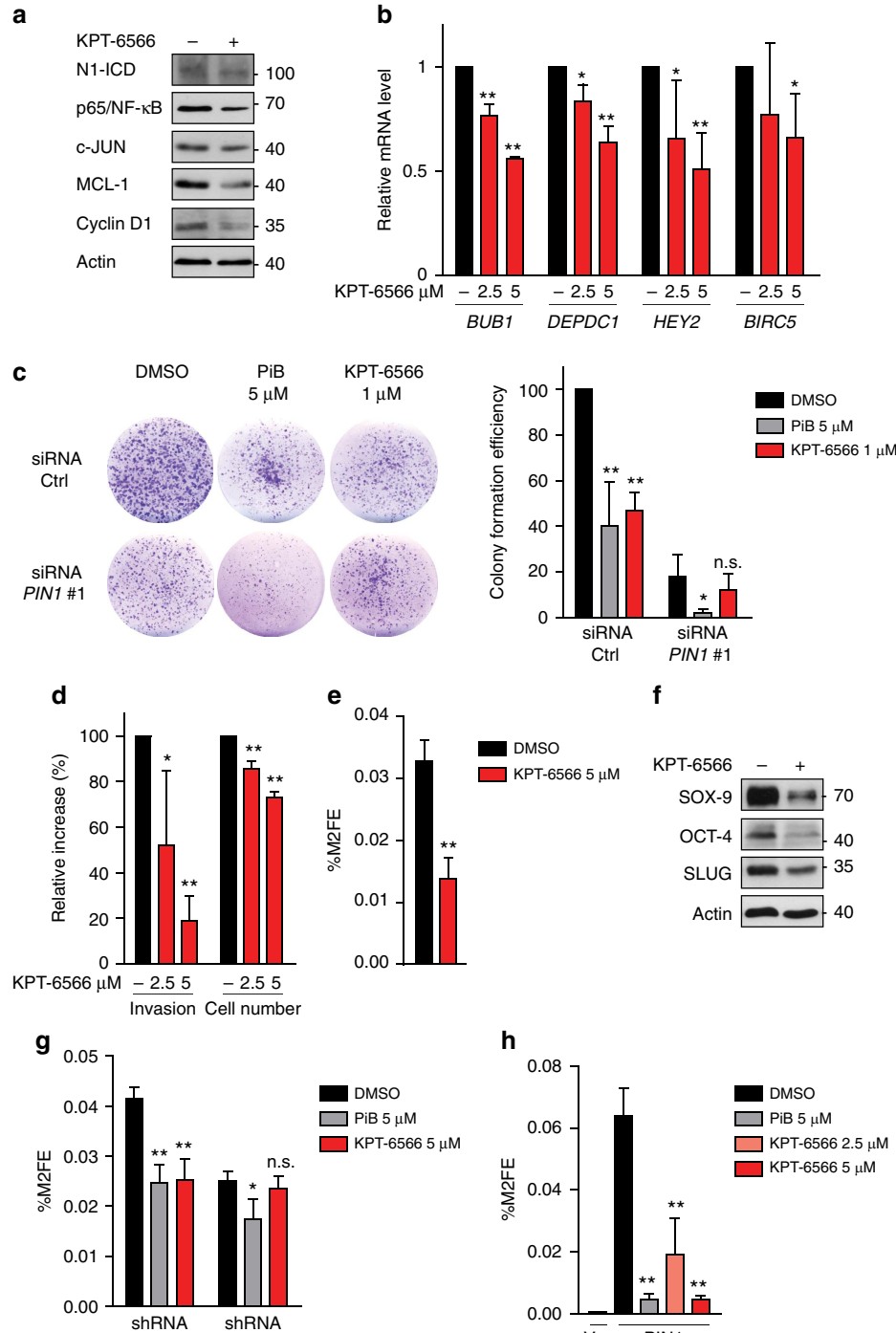

**Figure 3 | KPT-6566 interferes with PIN1 oncogenic functions.** (**a**) Immunoblotting of PIN1 client proteins expressed in MDA-MB-231 breast cancer cells treated with 5 µM KPT-6566 (+) or DMSO (−) for 48 h. (**b**) Quantitative RT-PCR analysis of mut-p53 and NOTCH1 target genes (*BUB1*, *DEPDC1* and *HEY2*, *BIRC5*, respectively) in MDA-MB-231 cells treated with indicated concentrations of KPT-6566 or DMSO (−) for 48 h. (**c**) Left, representative pictures of MDA-MB-231 colonies in the indicated experimental conditions. Right, histogram showing colony formation efficiency of MDA-MB-231 cells in the indicated experimental conditions. Cells were transfected with control siRNA (siRNA Ctrl) or with *PIN1* siRNA#1. After 24 h cells were trypsinized, plated for colony forming assay and treated with 5 µM PiB, 1 µM KPT-6566 or DMSO every two days. Colonies ≥50 pixels were counted 10 days after seeding using ImageJ software. (**d**) Histogram showing invasive ability and proliferation (cell number) of MDA-MB-231 breast cancer cells plated in Matrigel-coated Boyden chambers in the indicated experimental conditions for 20 h. (**e**) Histogram showing percentage of secondary mammosphere formation efficiency (%M2FE) of MDA-MB-231 cells in the indicated experimental conditions. (**f**) Immunoblotting of the indicated proteins expressed in MDA-MB-231 breast cancer cells treated with 5 µM KPT-6566 (+) or DMSO (−) for 48 h. (**g**) %M2FE of MCF10AT1 cells with stable control- (shRNA Ctrl) or *PIN1* silencing (shRNA *PIN1*), treated as indicated. (**h**) %M2FE of MCF10A cells transduced with empty- (Vec) or HA-PIN1-expressing vectors (PIN1), treated as indicated. (**a,f**) Actin levels are reported as loading control; size markers are indicated. Data shown in **b**–**e**,**g**,**h** are the means ± s.d. of *n* = 3 independent experiments, \*P < 0.05, \*\*P < 0.01, n.s. not significant; two-tailed Student's *t*-test.

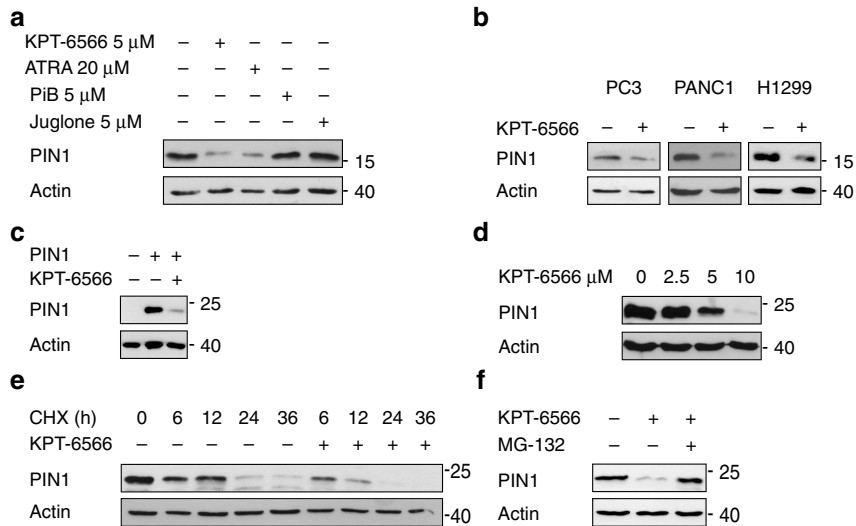

**Figure 4 | Interaction of KPT-6566 with PIN1 promotes its structural change and degradation.** Immunoblotting of the indicated proteins in cell lysates from (**a**) MDA-MB-231 cells treated with the indicated compounds (+) or DMSO (−) for 48 h; (**b**) PC3, PANC1 and H1299 cells treated with 5 μM KPT-6566 (+) or DMSO (−) for 48 h; (**c**) *PIN1* KO MDA-MB-231 cells transduced with empty (−) or PIN1 vectors (+) treated with 5 μM KPT-6566 (+) or DMSO (−) for 48 h; (**d**) *PIN1* KO MDA-MB-231 cells reconstituted with HA-PIN1, treated with increasing amounts of KPT-6566 or DMSO; (**e**) *PIN1* KO MDA-MB-231 cells reconstituted with HA-PIN1, treated with 5 μM KPT-6566 (+) or DMSO (−) followed by cycloheximide (CHX) chase for the indicated hours; (**f**) *PIN1* KO MDA-MB-231 cells reconstituted with HA-PIN1, treated with 5 μM KPT-6566 (+), 10 μM MG132 (+) or DMSO (−) for 16 h. **a–f** Actin levels are reported as loading control; size markers are indicated.

compound. To confirm this finding, we transduced MCF10A cells with empty- or HA-tagged PIN1-expressing vectors, and tested their M2FE. As already known, MCF10A cells per se have a very low M2FE, but this feature can be enhanced by PIN1 overexpression, which also induces an enrichment of the CD44[high]/CD24[low] cell population associated with stemness traits[7,35]. This PIN1-dependent gain of function was abolished by PiB and by KPT-6566 in a dose-dependent manner (Fig. 3h, Supplementary Fig. 3l,m and Supplementary Table 6).

**KPT-6566 promotes degradation of PIN1.** Slowly dissociating drugs, such as high affinity or covalent binders that cause structural changes of their target, have been shown to promote target degradation[36]. Notably, we consistently observed that KPT-6566 treatment of MEFs and human cancer cells caused a decrease of endogenous PIN1 levels (Figs 2b and 4a). *PIN1* promoter activity or *PIN1* mRNA levels were only slightly reduced following KPT-6566 treatment in MDA-MB-231 cells (Supplementary Fig. 4a,b), likely due to the disruption of the positive feed-forward loop existing between PIN1 and its transcription factors[27,29]. We thus hypothesized that the effect of KPT-6566 in reducing PIN1 levels mainly occurred through protein degradation. The observed decrease in PIN1 protein was similar to that obtained upon treatment with 20 μM all-*trans*-retinoic acid (ATRA), another PIN1 inhibitor shown to target PIN1 for degradation[12]. Instead, other PIN1 inhibitors, like Juglone or PiB, did not impact on PIN1 protein levels (Fig. 4a), as already observed by us and by others[7,27,37]. Moreover, upon KPT-6566 treatment, same effects were observed on endogenous PIN1 in cancer cell lines of different origins (Fig. 4b), and on HA-tagged PIN1 protein ectopically expressed in *PIN1* knockout MDA-MB-231 cells (Fig. 4c). KPT-6566 affected PIN1 levels in dose- and time-dependent manners (Fig. 4d,e), and by promoting PIN1 proteasomal degradation, since addition of MG132, a proteasome inhibitor, rescued the levels of both overexpressed and endogenous PIN1 (Fig. 4f and Supplementary Fig. 4c).

This indicates that the covalent modification of PIN1 induced by KPT-6566 elicits structural changes leading to PIN1 degradation.

**KPT-6566 elicits cellular responses to oxidative stress.** MS data suggest the transfer of the sulfanyl-acetate moiety of KPT-6566 to PIN1 C113 as a specific mode of action of this compound. Hence, besides irreversibly blocking the PIN1 active site, a remnant chemical species originates from KPT-6566 as a result of the KPT-6566-PIN1 interaction. This species can be released in the cellular compartment and interact with different cellular molecules. We hypothesized that this byproduct is 4-*tert*-butyl-N-(4-oxonaphthalen-1(4H)-ylidene)benzene-1-sulfonamide (hereinafter referred to as KPT-6566-B (2), Supplementary Methods). Containing a quinone-mimic substructure with the potential to generate reactive oxygen species, both KPT-6566 and KPT-6566-B may generate $H_2O_2$ in cells and induce oxidative stress[38]. In addition, KPT-6566-B has also features of a highly reactive electrophile that can engage different nucleophilic species in the cell and form DNA adducts[39] (Supplementary Fig. 5a).

To expand our observations regarding the action of KPT-6566 in cells, we compared the gene expression profiles of MDA-MB-231 cells treated with KPT-6566 or with *PIN1* RNAi, using an Illumina microarray platform. Treatments with DMSO and with control RNAi, respectively, were carried out as internal controls. Unsupervised hierarchical cluster analysis of the expression profiles of KPT-6566 and DMSO treated cells unveiled 499 significantly modulated genes. In addition, 832 genes were differentially expressed between *PIN1*- and control RNAi treated cells. Comparison of the two lists of genes revealed similarities between KPT-6566 and *PIN1* RNAi treated cells. Indeed, 34% of all genes downregulated by KPT-6566 were also downregulated after *PIN1* silencing (Fig. 5a and Supplementary Table 7). However, KPT-6566 treatment also induced a peculiar transcriptional program, since among its upregulated genes only 8% overlapped with those upregulated upon PIN1 silencing

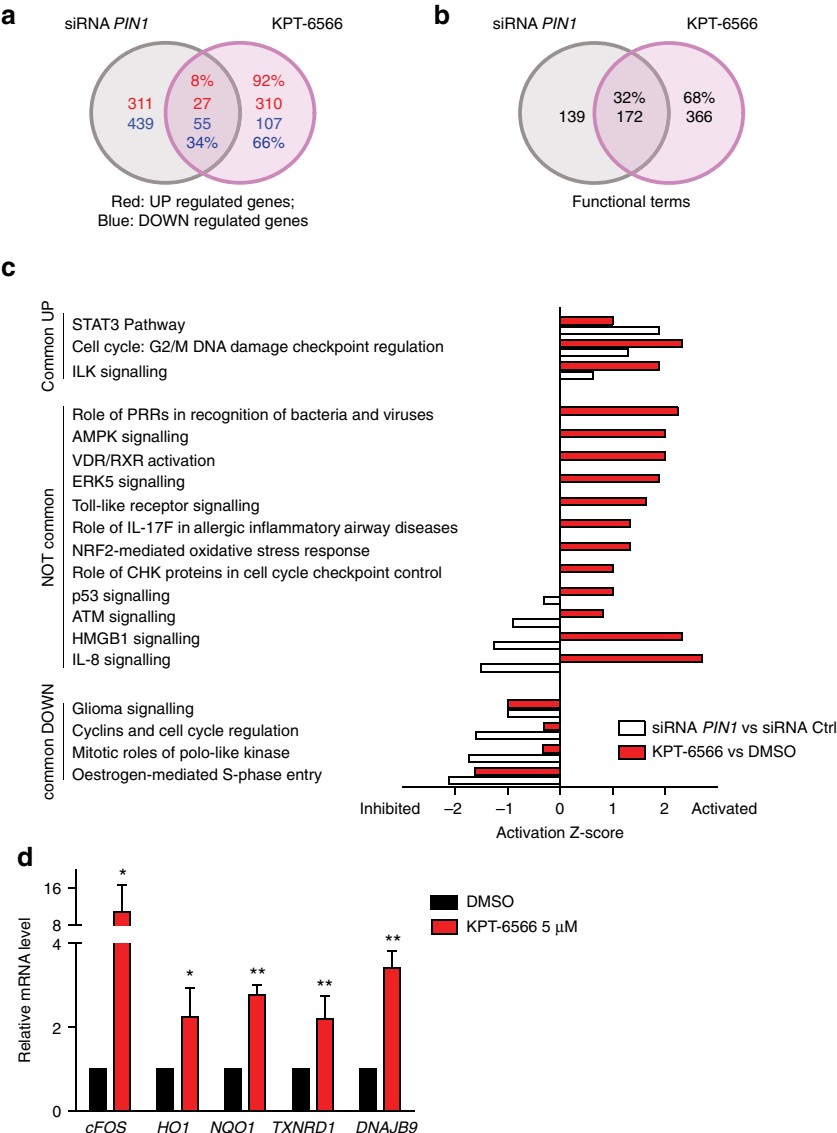

**Figure 5 | Global transcriptional effects of KPT-6566.** (**a**) Venn diagram of the Illumina microarray analysis with common and differentially up- and down-regulated genes of MDA-MB-231 cells treated with siRNA *PIN1*#1 vs. KPT-6566, using a cutoff of 0.75 logFC and *P*-value ≤ 0.05. (**b**) Venn diagram of the Gene Ontology (GO) analysis with common and differential enrichment for functional terms of MDA-MB-231 cells treated with siRNA *PIN1* vs. KPT-6566. (**c**) Histogram indicating comparison of IPA analysis of genes regulated by *PIN1* siRNA (white) or KPT-6566 (red) in function of their activation *Z*-scores. (**d**) Quantitative RT-PCR analysis of NRF2 pathway members in MDA-MB-231 cells treated with the indicated compounds for 48 h. Data are indicated as means ± s.d. of *n* = 3 independent experiments, *\*P* < 0.05, *\*\*P* < 0.01; two-tailed Student's *t*-test.

(Fig. 5a and Supplementary Table 8). To obtain more information about the cellular processes affected by KPT-6566, we performed Gene Ontology and Ingenuity Pathway Analysis (IPA). Both analyses revealed a common enrichment for biological themes and pathways connected to cell cycle and cell proliferation, as expected for compounds targeting PIN1 in cancer cells[4], supporting the notion that part of the effect of KPT-6566 on gene transcription is mediated by PIN1 inhibition (Fig. 5b,c and Supplementary Table 9). We confirmed this result by RT-qPCR analysis of the expression of selected genes related to cell cycle (Supplementary Fig. 5b and Supplementary Table 10). Strikingly, IPA analysis revealed that, besides PIN1 inhibition, KPT-6566 elicits additional effects, in particular perturbing pathways related to inflammation and oxidative stress response (Fig. 5c). These pathways were predicted to be activated by KPT-6566 but not modulated by *PIN1* silencing. In line with the reactive nature of

KPT-6566 and KPT-6566-B, the NRF2-mediated oxidative stress response, which physiologically represents one of the most important, intracellular, antioxidant mechanisms[40], emerged among others as a pathway differentially regulated by KPT-6566. We analysed several genes of this pathway and the expression data demonstrated a good correlation with microarray analyses (Fig. 5d). Altogether these analyses support the notion that KPT-6566 acts on one hand as a PIN1 inhibitor, and on the other as an inducer of cellular stress responses.

Given this premise and considering the induction of the NRF2-dependent antioxidant response by KPT-6566, we analysed ROS levels by CellROX FACS analyses in MDA-MB-231 and PC3 cells upon KPT-6566 administration. This analysis showed that treatment with KPT-6566 increased endogenous ROS levels, and this effect was drastically reduced by adding *N*-Acetyl-Cysteine (NAC), a ROS scavenger and thiol group donor[41]

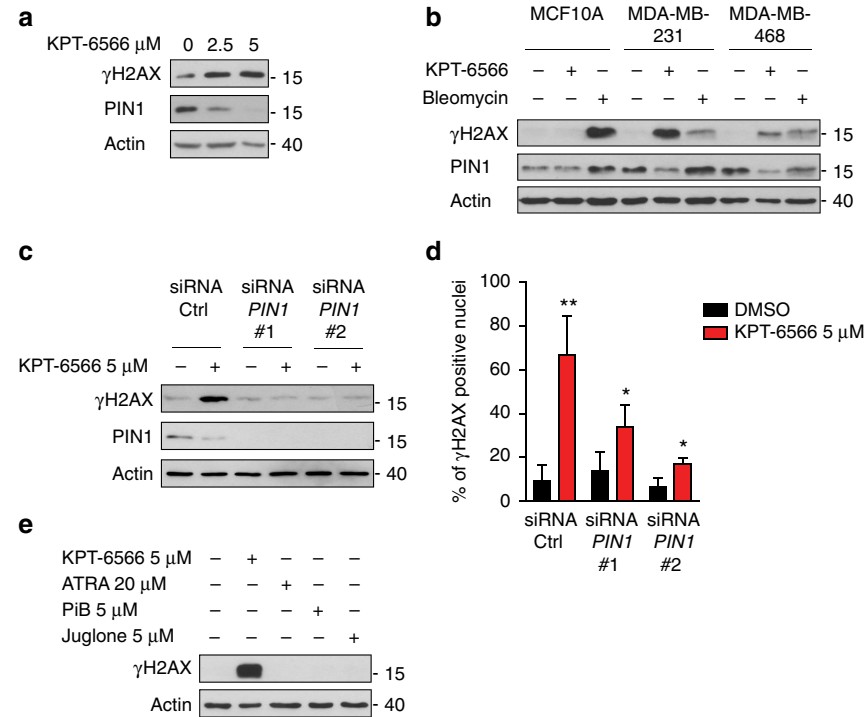

**Figure 6 | KPT-6566 induces DNA damage in a PIN1-dependent manner.** Immunoblotting of the indicated proteins from (**a**) MDA-MB-231 cells treated with increasing amounts of KPT-6566 or DMSO for 48 h; (**b**) MCF10A, MDA-MB-231 and MDA-MB-468 cells treated with 5 µM KPT-6566 ( + ), 10 µM Bleomycin ( + ) or DMSO ( − ) for 48 h; (**c**) MDA-MB-231 cells transfected with control- (siRNA Ctrl) or two different *PIN1* siRNAs (siRNA *PIN1*#1 and *PIN1*#2) and treated as indicated for 24 h. (**d**) Histogram representing the percentage of γH2AX positive nuclei of MDA-MB-231 cells treated as in **c**. (**e**) Immunoblotting of the indicated proteins from MDA-MB-231 cells treated with the indicated compounds ( + ) or DMSO (–) for 48 h. **a**–**c**,**e** Actin levels are reported as loading control; size markers are indicated. Data shown in (**d**) are the means ± s.d. of *n* = 3 independent experiments, \**P* < 0.05, \*\**P* < 0.01; two-tailed Student's *t*-test.

(Supplementary Fig. 5c). A similar result was obtained by treating MDA-MB-231 cells with KPT-6566-B (Supplementary Fig. 5d).

To investigate if KPT-6566 treatment elicits ROS formation also in normal cells, we performed CellROX analyses in MCF10A cells and observed an increase also in this cell type (Supplementary Fig. 5c). However, in MCF10A cells the amount of ROS was significantly lower as compared to cancer cells and KPT-6566 treatment did not alter cell viability (Fig. 2g).

**KPT-6566 induces DNA damage in a PIN1-dependent manner.** In addition to ROS production elicited by both KPT-6566 and KPT-6566-B, KPT-6566-B might be also involved in the formation of DNA adducts due to its unsubstituted quinone-mimic substructure. DNA adducts or ROS induced DNA damage are either repaired and/or can be converted into double strand breaks (DSBs)[39,42]. It is thus conceivable that treatment with KPT-6566, once converted to KPT-6566-B, could induce DNA damage thus evoking a DNA damage response (DDR)[43]. To test this hypothesis, we analysed histone H2AX Ser139 phosphorylation (γH2AX), a marker of DNA damage[43], in MDA-MB-231 cells treated with KPT-6566 or KPT-6566-B. Both compounds, indeed, elicited a dose-dependent increase of H2AX phosphorylation (Fig. 6a and Supplementary Fig. 6a).

KPT-6566 treatment caused H2AX phosphorylation also in other cancer cell lines, while normal immortalized MCF10A breast cells did not show signs of DDR. Treatment with Bleomycin, a radiomimetic known to induce DSBs, caused H2AX phosphorylation in all cell lines, indicating that the DDR mechanisms were intact (Fig. 6b and Supplementary Fig. 6b).

We next asked whether KPT-6566 treatment caused DNA damage in a PIN1-dependent manner. As show in Fig. 6c,d and Supplementary Fig. 6c,d, PIN1 was required for the DNA damaging activity of KPT-6566, but not of KPT-6566-B. In fact, KPT-6566 treatment in MDA-MB-231 cells caused H2AX phosphorylation and γ-H2AX-positive foci formation in the majority of cells. However, while KPT-6566 effect was almost undetectable in *PIN1* silenced cells (Fig. 6c,d and Supplementary Fig. 6c), KPT-6566-B DNA damaging activity was independent from PIN1, since the compound elicited H2AX phosphorylation to a same extent in control and *PIN1* silenced MDA-MB-231 cells (Supplementary Fig. 6d). Accordingly, KPT-6566-B did not cause any decrease in PIN1 levels in MDA-MB-231 cells (Supplementary Fig. 6a,d).

Notably, other PIN1 inhibitors, such as ATRA, PiB or Juglone, were not able to induce H2AX phosphorylation at the concentrations generally used to inhibit PIN1 (Fig. 6e). These results indicate that the presence of PIN1 is required for the DNA damaging activity of KPT-6566, and demonstrate that KPT-6566 treatment elicits a PIN1-mediated intracellular release of KPT-6566-B, causing DNA damage. This effect was partially rescued by treatment with the highly efficient ROS scavenger melatonin[44] (Supplementary Fig. 6e), suggesting that besides ROS production, the release of KPT-6566-B might contribute to DNA damage through formation of DNA adducts.

**KPT-6566 induces cell death in cancer cells.** Upon KPT-6566 administration, we observed DNA damage and killing effects preferentially in cancer cells (Figs 2e,g and 6b). Tumour cells are sensitive to DNA damage and depletion of antioxidant reservoirs,

and, over a certain threshold, stress overload may lead to cancer cell death[45,46]. Hence, we hypothesized that, upon KPT-6566 treatment, the observed acute increase of ROS and DNA damage along with the effects of PIN1 inhibition, would represent the *coup de grâce* leading to proliferation arrest and cell death. To assess whether KPT-6566 treatment impaired cancer cells viability by inducing both proliferation arrest and cell death, we performed a Trypan blue assay in control- or *PIN1* silenced MDA-MB-231 cells treated with DMSO or KPT-6566 (Supplementary Fig. 7a). As expected, *PIN1* silencing caused growth arrest[27], but not cell death. KPT-6566 treatment caused both a decrease in cell number and an increase of the number of dead cells, indicating that, in addition to the proliferation arrest associated with PIN1 inhibition, KPT-6566 also induced cell death. Propidium Iodide/Annexin V FACS analyses confirmed these results, indicating that KPT-6566 induced cell death with traits of late apoptosis and necrosis (Fig. 7a and Supplementary Fig. 7b). We investigated this feature in cell lines from breast, prostate, lung and pancreatic tumours by evaluating the presence of cell death markers (PARP cleavage, cleaved Caspase-3, secreted HMGB1)[47] by western blot analysis (Fig. 7b and Supplementary Fig. 7c). In line with the previous observations, all tested cell lines showed markers of both apoptotic and necrotic cell death, supporting a widespread killing activity of KPT-6566 towards cancer cells.

Finally, we evaluated the effect of KPT-6566 *in vivo* and as a first step we tested its general toxicity in nude mice. At high doses of the compound (60 or 90 mg kg$^{-1}$ injected intravenously) a strong phlebitis was observed at the site of injection. Chronic intraperitoneal (i.p.) administrations of 30 and 45 mg kg$^{-1}$ of KPT-6566 were better tolerated, causing only local, non life-threatening toxicity at the site of inoculation, where granulation and fibrotic thickening of the peritoneal wall were observed (personal observations). On the basis of this result, we tested the effect of a daily administration of 5 mg kg$^{-1}$ i.p. of KPT-6566 in a lung colonization assay. To this aim, we performed tail vein injection of MDA-MB-231 cells in nude mice (15 animals). The day after cancer cell injection, mice were randomized in two groups to be treated daily with either KPT-6566 or the vehicle. Twenty-seven days after cancer cell inoculation, control mice began to show signs of distress, as determined by body weight loss (Supplementary Fig. 7d). Mice were killed and lungs were extracted and analysed for metastatic growth by both metastatic area determination and organ weight. The metastatic growth in KPT-6566 treated animals was significantly reduced compared to controls (Fig. 7c, and Supplementary Fig. 7e). Interestingly, post mortem morphologic analyses of vital organs did not reveal any sign of local or systemic/ organ toxicity.

## Discussion

The prolyl isomerase PIN1 represents a critical player in several signalling circuitries characterizing both CSCs and non-CSC tumour cells. As a consequence of this function, PIN1 inhibition causes the collapse of numerous oncogenic pathways at the same time, making this isomerase an attractive drug target for the development of treatments against aggressive and drug-resistant cancers. Despite considerable efforts, however, poor success has been reached so far: the available PIN1 inhibitors either lack the required specificity and/or potency, or cannot efficiently enter cells to inhibit PIN1 function *in vivo*.

Through a mechanism-based screening, we have identified a novel PIN1 small molecule inhibitor (KPT-6566). KPT-6566 is characterized by a unique mechanism of action, combining PIN1 inhibition with the release of a cytotoxic molecule. KPT-6566 blocks PIN1 catalytic activity by covalent interaction and elicits PIN1 proteasome-dependent degradation. Moreover it represents a selective inhibitor of PIN1, as it does not interact with other thiol-containing PPIases.

Several PIN1 inhibitors have been identified so far with both covalent and non-covalent mechanisms of action[5,14]. Among them, ATRA, Juglone and PiB, have shown activity in cells and in mouse models[12,48,49]. Our study indicates that KPT-6566 has a number of advantages over these PIN1 inhibitors. As KPT-6566, Juglone covalently interacts with PIN1 catalytic domain. However, Juglone has a relatively simple structure that affects its specificity. Indeed, Juglone has several off-targets and manifests PIN1 independent activities[14,50]. Unlike Juglone or PiB, which exerts its effects also through Parvulin 14 inhibition, KPT-6566 is highly specific towards PIN1 and mainly ineffective in cells deprived of PIN1. Like KPT-6566, also ATRA promotes PIN1 degradation. Despite this and the advantage of being an FDA approved drug, ATRA showed less potency towards PIN1 than KPT-6566, likely due to its non-covalent mechanism of action and its very short half-life[12].

Most of all, however, KPT-6566 represents a one of a kind PIN1 inhibitor because it associates a highly specific PIN1 inhibitory activity with the release of a reactive quinone-mimicking byproduct that acts downstream of PIN1 and generates DNA damage and elicits cancer cell death. Indeed, as a PIN1 inhibitor and a ROS producing and DNA damaging agent, KPT-6566 exerts both cytostatic and cytotoxic effects (Fig. 7d). As a consequence of PIN1 inhibition and degradation, KPT-6566 reduces levels and/or activity of several PIN1 client proteins, such as Cyclin D1, c-JUN, MCL-1, N1-ICD, and mut-p53. Although the effects on single client proteins are only moderate, the simultaneous impairment of multiple PIN1 targets may account for the observed strong suppression of all tested, cancer-related phenotypes depending on PIN1 function, namely proliferation, colony formation, invasion and CSC maintenance. As a consequence of the release of a DNA damaging byproduct and of ROS generation, KPT-6566 induces a significant increase of DNA damage and cell death specifically in cancer cells, while only a slight increase of ROS levels in normal cells. Normal cells express lower levels of PIN1 than cancer cells. This may account for a slower kinetic of KPT-6566-B and DNA damage production in these cells and for their limited sensitivity to KPT-6566 treatment. Moreover, normal cells are exposed to low levels of oxidative stress. Cancer cells, instead, are characterized by increased DNA damage and oxidative stress that make them more vulnerable to agents causing further accumulation of ROS and DNA damage[45,46]. Indeed, upon KPT-6566 treatment, acute ROS overload along with a conspicuous increase of DNA damage due to KPT-6566-B production causes cancer cells death. Consistent with the pharmacological, biochemical and anti-cancer properties observed *in vitro*, KPT-6566 chronic administration in mice effectively reduced MDA-MB-231 lung colonization without major toxicity. Further studies are required to dissect the mechanisms involved in DNA damaging activity of KPT-6566-B, such as adducts formation.

The mechanism of action of KPT-6566 meets the criteria of modern rational drug design envisaging targeted delivery of drug conjugates that release their cytotoxic counterparts specifically in cancer cells[51]. Although additional structural modifications might further improve KPT-6566 drug-likeness, its chemical features and selectivity make it already an attractive molecule to be developed for a potential use as an anti-cancer drug in humans.

## Methods

**Virtual screening.** A library of commercial compounds was generated using a drug like collection obtained from Asienx (www.asienx.com), Maybridge (www.maybridge.com), Bionet (www.keyorganics.co.uk), Specs (www.specs.net),

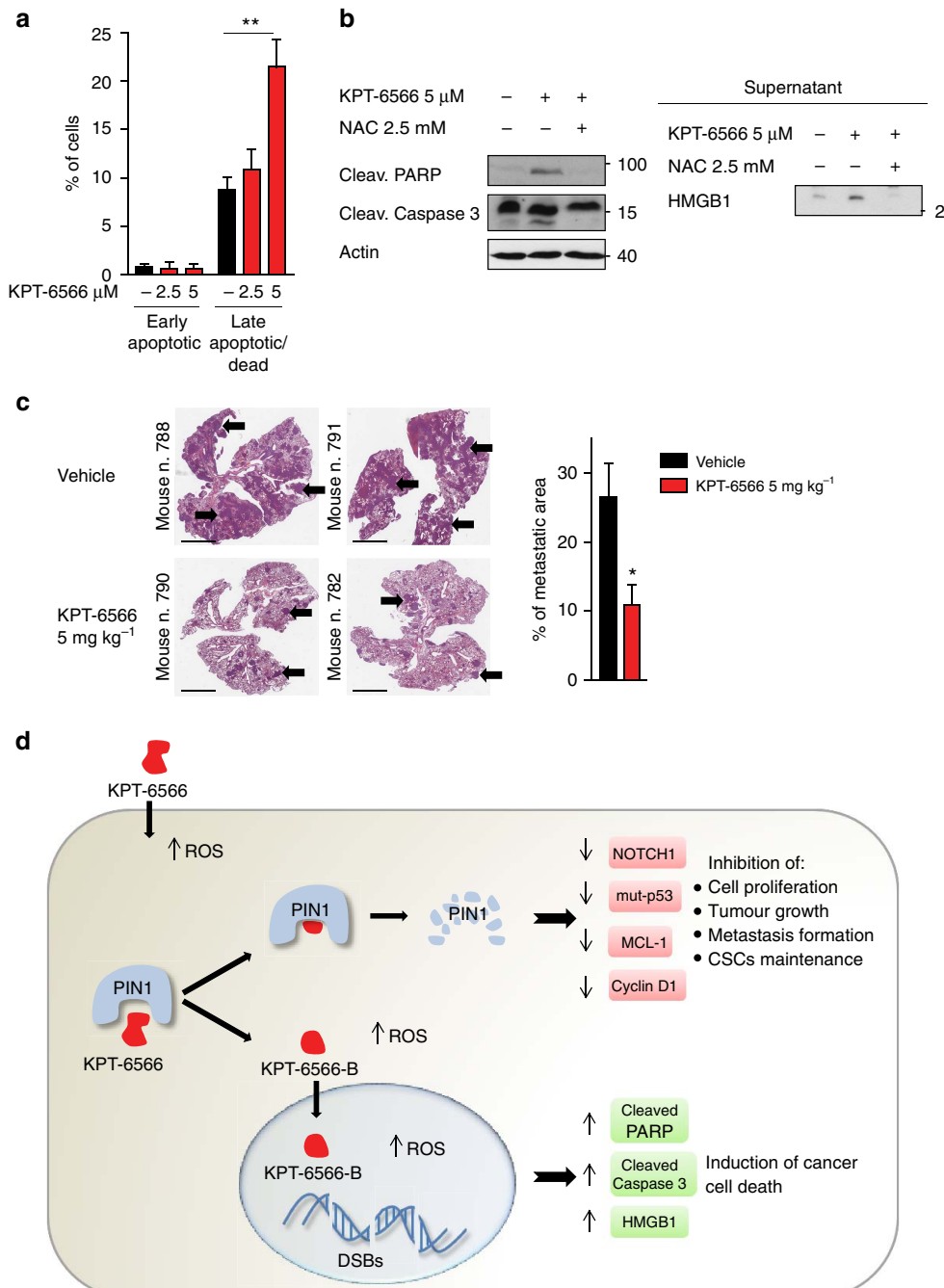

**Figure 7 | KPT-6566 induces cell death of cancer cells and reduces metastasis growth *in vivo*.** (**a**) Histogram representing respectively the percentage of early apoptotic and late apoptotic/dead MDA-MB-231 cells in the indicated experimental conditions. Results are indicated as means ± s.d. of $n = 3$ independent experiments, **$P < 0.01$; two-tailed Student's $t$-test. (**b**) Left, immunoblotting of the indicated proteins of MDA-MB-231 cell lysates untreated ( − ) or treated ( + ) with the indicated compounds. Right, immunoblotting of HMGB1 from supernatant of the same cells as in the left panel; actin levels are reported as loading control; size markers are indicated. (**c**) Left, hematoxylin and eosin staining of representative sections of entire pulmonary lobes from mice inoculated with MDA-MB-231 cells and treated with vehicle ($n = 8$) or KPT-6566 ($n = 7$); arrows indicate representative metastases. Scale bar, 4 mm. Right, computer-aided assessment of percentage of lung tissue area occupied by metastases in the indicated conditions. Data are reported in histograms as means ± s.e.m.; *$P < 0.05$; two-tailed Mann–Whitney test. (**d**) Proposed model of KPT-6566 mechanism of action. After entering the cell, KPT-6566 binds to PIN1 and elicits a cytostatic effect associated to covalent inhibition and degradation of PIN1 with consequent decrease of oncogenic circuitries. In addition, KPT-6566 might induce intracellular ROS production (upper part). After reaction with PIN1, KPT-6566 has a simultaneous cytotoxic effect releasing KPT-6566-B, which generates further ROS, induces DNA damage and cancer cell death (lower part).

Chembridge (www.chembridge.com), ChemDiv (www.chemdiv.com) and Enamine (www.enamine.net). Virtual Screening of this collection was performed using a covalent docking approach suitable for large-scale virtual screening (VS) called CovDock-VS method previously described[17] on the structure of the catalytic pocket of PIN1. Briefly, compounds were prepared using the Virtual Screening Workflow (VSW) ligand preparation tab in Maestro (Maestro v9.2. (2011)

Schrödinger, Inc., Portland). 'Regularize input geometries' was applied and ionization states and tautomers were determined by the ionizer at a pH 7.4. Compounds were subsequently filtered in Canvas using the following chemical property ranges that correspond to the Lipinski's rule of five for enhanced drug-likeness: MW 300–550, hydrogen-bond donors 0-5, hydrogen-bond acceptors 1–10, Rotatable bonds < 10, AlogP < 5.5, Total charge -1à1, PSA < 140. The PIN1

structure (PDBID: 2XPB)[16] was used for structure-based screening following protein preparation in Maestro (Maestro v9.2. (2011) Schrödinger, Inc., Portland). In short, this included assignment of bond orders for ligands, addition of hydrogen atoms, optimization of the hydrogen bonding network and a restrained minimization. All default options were used.

**Cytotoxicity tests.** Mouse fibroblast 3T3 and human malignant melanoma A375 cells were resuspended in DMEM cell culture medium containing 10% fetal bovine serum (Gibco) and antibiotics (penicillin and streptomycin, Gibco). Hundred microlitres per well was transferred to the assay plate (5,700 or 4,700 cells per well for 3T3 and A375, respectively) and incubated overnight in a humidified incubator at 37 °C and 5% $CO_2$. KPT compounds were serially diluted in assay media (9 serial 1:3 dilutions) and 100 μl per well were added to the assay plate. Final assay volume of each well was 200 μl, containing KPT compounds, starting from 30 μM concentration downwards. After 70 h of incubation 20 μl of Promega Substrate CellTiter 96 AQ$_{ueous}$ One Solution Cell Proliferation Assay Reagent were added to each well and, according to the manufacturer's procedures, after short incubation at room temperature in a humidified chamber protected from light, absorbance was read at 490 nm. Means of data collected in duplicate were plotted on a logarithmic scale and IC50 values were obtained.

**Protease coupled isomerization assay.** Recombinant PIN1 protein, 0.5 nM, was incubated in a 96-multi-well plate for 3 h at 0 °C in 80 μl HEPES buffer pH = 7.8 in presence of 30 μM KPT compounds, or DMSO. In parallel, 0.5 nM bovine serum albumin (BSA, PanReac Applichem) or PIN1 S67E were incubated with DMSO as above. Immediately before measurement 15 μl of Trypsin 50 mg ml$^{-1}$ (Sigma) and 5 μl of Suc-AEPF-MCA peptide 100 μg ml$^{-1}$ (ChemDiv) in TFE LiCl 480 mM were added and the solutions mixed. Hydrolysis of the substrate was monitored by measuring the fluorescence of the released MCA exciting at 370 nm and detecting at 440 nm every 5 s for 5 min using an EnSpire Perkin Elmer multiplate reader. Normalized fluorescence data were converted in Suc-AEPF-MCA concentration values that were used to calculate the PPIase activity $K$ following the method described in Kullertz et al.[52] using the value of BSA ($k_0$) as reference of an uncatalysed reaction. The same procedure was followed for GST-FKBP4 and GST-PPIA PPIase assays using Suc-ALPF-MCA and Suc-AAPF-MCA peptides (Bachem), respectively. Proteins were incubated with Cyclosporin A (Calbiochem #239835), FK506 (Calbiochem #342500) or KPT-6566. Oligonucleotides for construction of expressing vectors and recombinant protein production are detailed in Supplementary Table 11 and in the Supplementary Methods section.

**Determination of IC50 and inactivation constants in PPIase assays.** For IC50 determination, human recombinant PIN1 was preincubated with different concentrations of KPT-6566 and PPIase activity was measured after 180 min. $K$ values of PPIase activity were plotted against inhibitor concentration in a semi-logarithmic plot. IC50 values were calculated using log (inhibitor) vs. normalized response function of GraphPad Prism software (GraphPad Software, La Jolla, CA). For inactivation constants determination, human recombinant PIN1 was preincubated with different concentrations of KPT-6566 and PPIase activity was measured after 0, 60, 120 or 180 min. The logarithm of the percentage of the remaining isomerization activity was plotted against preincubation time, yielding a semi-logarithmic plot. The observed rate constant for inhibition, $k_{obs}$, at each concentration was determined from the slope of the semi-logarithmic plot of inhibition vs. time. The $k_{obs}$ values were re-plotted against inhibitor concentration, and fitted to a hyperbolic equation (One site binding-hyperbola; GraphPad Software), according to the equation $k_{obs} = k_{inact}[I]/(K_i + [I])$, to obtain values for $K_i$ and $k_{inact}$. $k_2$ is the rate constant that defines the maximal rate of inactive enzyme formation, I is the initial concentration of the inhibitor and $K_i$ is the inhibitor concentration when $k_{obs} = k_{inact}/2$. The $k_{inact}/K_i$ ratio represents the second-order rate constant for the reaction of the inhibitor with the target.

**LC/MS analysis.** For LC-MS analyses KPT-6566 was dissolved in DMSO at a concentration of 10 mM. PIN1 samples were prepared in 30 mM HEPES buffer pH 7.9. PIN1 protein (10 μg) at a concentration of 9 μM was incubated with threefold excess compound KPT-6566, in 100 mM HEPES buffer pH 7.9 for 1 h in ice. The reaction was stopped by adding 1 μl of 10% v/v TFA and the samples were injected in the HPLC/MS instrument. WT PIN1 or the cysteine mutant C113A, incubated with KPT-6566, KPT-6566 in combination with DTT, or in 2.7% DMSO were analysed for full MW determination by liquid chromatography/electrospray ionization mass spectrometry (LC/ESI-MS) using a 1100 HPLC apparatus coupled through an API-ESI source to a 1946 MSD single quadrupole MS detector, both from Agilent (Palo Alto, CA, USA). A RP Poroshell 300SB-C3 column (2.1 mm ID × 75 mm, 5 μm) was used and proteins were eluted applying a 0.05% TFA/acetonitrile gradient. Acquired MS spectra were deconvoluted using the ChemStation deconvolution software package (Agilent).

**Protein digestion and MALDI-ToF-ToF MS and MS/MS analysis.** PIN1 protein was digested adding ProteaseMAX Surfactant Trypsin Enhancer (Promega Corporation) (0.01% v/v final concentration) and 1 mg of TPCK trypsin to the

reaction solution. The mixture was incubated for 30 min at 37 °C. A volume of 0.3 μl of the samples were mixed with 0.3 μl of alpha-cyano-4-hydroxycinnamic acid (HCCA) matrix (20 mg in 1 ml of 50% acetonitrile, 0.1% TFA) loaded on the MALDI plate and analysed by 4800 MALDI ToF/ToF (ABI Sciex) in reflector conditions using optimized parameters. Peptides at $m/z$ possibly containing modified cysteine were subjected to MS/MS analysis. Obtained spectra were manually inspected to reconstruct the sequence.

**Molecular-docking study.** Computational studies were carried out using Schrödinger suite running on a Linux based customized work station with CentOS platform. Grid-Based Ligand Docking with Energetics (Glide) program was used to analyse the binding conformations of KPT-6566 and PIN1 (ref. 21). Three-dimensional (3D) structures of PIN1 (PDBID: 2XPB)[16] were retrieved from the Protein Data Bank (www.rcsb.org). The protein structures were organized using the Protein Preparation Wizard, whereas the 3D structure of the ligand was optimized using LigPrep module and the partial charges were ascribed using Optimized Potentials for Liquid Simulations (OPLS3) force-field[53]. Hydrogen atoms were added to the protein structure to pH 7.0 considering the appropriate ionization states for both the acidic and basic amino acid residues. The binding pocket of PIN1 was defined by a 10 × 10 × 10 Å box centred on the central position of C113. 'Extra-Precision' (XP) mode of Glide application of Schrödinger suite helped to measure the binding affinity of proteins to ligand docking. The protocol facilitates flexible ligand docking within the rigid receptor and the best ligand pose with reference to the protein was chosen based on the grading obtained by Glide Score[23].

**Cell lines and treatments.** Unless otherwise stated, all cell lines were purchased from ATCC or obtained from other laboratories cooperating on the project. Human cell lines were subjected to STR genotyping with PowerPlex 18D System and their identity was confirmed comparing the results to reference ATCC, DMSZ and JCRB databases and are not listed in the NCBI Biosample database of commonly misidentified cell lines. They were also routinely tested for absence of Mycoplasma infection by PCR/Immunofluorescence. HMEC are normal hTERT immortalized epithelial breast cells[54]. MCF10A are normal immortalized epithelial breast cells, MCF10AT1 are xenograft-passaged T24 H-Ras transformed MCF10A cells[28]. MDA-MB-231 and MDA-MB-468 are breast carcinoma cell lines, H1299 is a lung carcinoma cell line, PC3 is a prostatic carcinoma cell line, LNCaP is a prostatic carcinoma cell line derived from metastatic site, PANC1 is a pancreatic carcinoma cell line, SKOV3 is a ovarian carcinoma cell line and HeLa is a human cervical carcinoma cell line. PIN1 knockout MDA-MB-231 cells were generated by CRISPR/Cas9 genome engineering technology, as described in Supplementary Method section. Immortalized $Pin1^{-/-}$ mouse fibroblasts were obtained by spontaneous immortalization from MEFs of C57BL6/129Sv mixed background and were already described[7]. MDA-MB-231, MDA-MB-468, PANC1, SKOV3, HeLa and MEFs were cultured in DMEM (BioWhittaker) supplemented with 10% fetal bovine serum (FBS, Gibco), 100 U ml$^{-1}$ penicillin and 100 μg ml$^{-1}$ streptomycin (Euroclone). H1299, PC3 and LNCaP cells were grown in RPMI (BioWhittaker) supplemented with 10% fetal bovine serum and penicillin/streptomycin. MCF10A and MCF10AT1 cells were maintained in DMEM:F12 Ham's (1:2) (BioWhittaker), supplemented with 5% horse serum (Gibco), 10 μg ml$^{-1}$ insulin (Sigma), 0.5 μg ml$^{-1}$ hydrocortisone (Sigma) and 20 ng ml$^{-1}$ EGF (Cell Guidance System). HMEC cells were maintained in MEBM (BioWhittaker), supplemented with 0.4% Bovine Pituitary Extract (Life Technologies), 5 μg ml$^{-1}$ insulin, 0.5 μg ml$^{-1}$, hydrocortisone and 10 ng ml$^{-1}$ EGF. Transient transfections were performed using standard procedures as already described[7,27]. siRNA sequences are listed in Supplementary Table 12. For creation of stable clones, a selection corresponding to the expressed vectors was applied for 2 weeks to transfected or infected cells at concentrations of 2 μg ml$^{-1}$ for puromycin (Sigma) and 5 μg ml$^{-1}$ for blasticidin (InvivoGen). KPT-6566 (10 mM) or KPT-6566-B (5 mM) dissolved in DMSO was used for cell treatments at the indicated concentrations. Control cells received DMSO at the same concentrations ranging from 0.05 to 0.01%. PPIase-Parvulin Inhibitor PiB (Calbiochem #529627), Juglone (Calbiochem #420120), ATRA (all-trans-retinoic acid, Sigma R2625), MG132 (Calbiochem #474790), NAC (N-Acetyl-L-Cysteine, Sigma A7250), Melatonin (Sigma M5250) and Bleomycin (EURO Nippon Kayaku GmbH, Sanofi Aventis), were resuspended as indicated in the respective datasheets.

**Immunoblotting and immunofluorescence.** For immunoblotting analyses protein lysates were loaded and separated in SDS-PAGE, followed by Western blotting on Nitrocellulose membranes (Amersham). Blocking was performed in Blotto-tween (PBS, 0.2% Tween-20, not fat dry milk 5%) or with TBST (0.2% Tween-20, Tris/HCl 25 mM pH 7.5) plus 5% BSA (PanReac Applichem) depending on the antibody, as described[55]. Immunoblot analyses were performed at least from three biological replicates. For immunofluorescence, after 48 h of treatment with the compounds, cells were fixed in 4% paraformaldehyde for 20 min, washed in PBS, permeabilized with Triton 0.1% for 5 min and blocked in PBS + FBS 1% + BSA 0.2% for 30 min. Antigen recognition was done by incubating primary antibody for 1 h at 37 °C and with goat anti-mouse Alexa Fluor 568 as secondary antibody for 30 min at 37 °C. Nuclei were counterstained with Hoechst 33342 (Life Technologies). Representative fluorescence images were taken with a × 630 magnification on a Leica DM4000B microscope equipped with a LEICA DFC420C

camera (Leica Microsystems S.r.l. Milan, Italy) and acquired with Leica Application Suite 2.5.0 R1 (Leica Microsystems).

**Quantitative real-time PCR (qRT-PCR) analysis.** Total RNA from cell lines was extracted with QIAzol Lysis Reagent (Qiagen) and cDNA was transcribed with QuantiTect (Qiagen) in accordance with manufacturer's protocols. cDNA was then amplified on a CFX96 Touch Real-Time PCR Detection System with SsoAdvanced SYBR Green Supermix (Biorad) and analysed with Biorad CFX Manager software. Expression levels are always given relative to histone *H3*. Oligonucleotide sequences are listed in Supplementary Table 10.

**Proliferation assays.** For proliferation curves, mouse embryo fibroblasts were seeded at a density of 5,000 cells per well in a 6-well plate and incubated for 24 h in 10% FBS-supplemented DMEM culture medium. Cells were then treated with KPT-6566 or DMSO every second day. The number of cells was counted every second day with hemocytometer after trypsin digestion.

For Trypan blue staining, cells were seeded at a density of 250,000 cells per well in a 6-well plate and incubated for 24 h in 10% FBS-supplemented DMEM culture medium. Cells were then treated with KPT-6566 or DMSO. After 24 h the number of cells was counted after trypsin digestion and staining with 0.2% Trypan Blue (Sigma).

**Viability assays.** For WST assay, MCF10A and MCF10AT1 cells were seeded in 96-multi-well plate at a concentration of 4,000 cells per well. Twenty-four hours after seeding, cells were treated with the compound or DMSO as a negative control. Forty-eight hours after compounds administration, growing medium was removed and WST-1 was added to the cells (Promega). Absorbance was read in an EnSpire Perkin Elmer multiplate reader. For ATPlite assay, cells were seeded in 96-multi-well view-plate at a concentration of 4,000 cells per well. Twenty-four hours after seeding, cells were treated with the compound or DMSO as a negative control. Forty-eight hours after compound administration, growing medium was removed and cells were lysed with 50 μl of ATPlite 1step substrate solution (Perkin Elmer), as described[56]. Luminescence was read in an EnSpire Perkin Elmer multiplate reader. IC50 values were calculated using log (inhibitor) vs. normalized response function of GraphPad Prism software.

**Colony formation assay.** Cells were seeded at a density of 5,000 (MDA-MB-231) or 7,500 cells (PC3) per 6 cm diameter plate and incubated for 24 h in 10% FBS-supplemented DMEM culture medium. Cells were then treated with KPT-6566, PiB or DMSO. After 10 days, cells were fixed in 4% formaldehyde and stained with Giemsa (Sigma) diluted solution 1:10 in water for 2 h. Colonies ≥50 pixels were counted using ImageJ[57] software after background subtraction.

**Invasion assay.** Invasion assays were performed by seeding cells at a density of 5,000 cells per well in 24-well PET inserts (8.0 μm pore size, Falcon) with matrigel-coated filters. Cells were treated for 20 h with KPT-6566 or DMSO and then invading cells were fixed, stained with 0.5% Crystal Violet (Sigma) and counted with a ×20 objective on CK30 Olympus optical microscope (Olympus Italia Srl, Milan, Italy).

**Mammosphere cultures.** To obtain mammospheres, cells from monolayer cultures were enzymatically disaggregated (0.05% trypsin-EDTA, Gibco) to a single-cell suspension, passed though a 40 μm cell strainer (BD Falcon), plated at clonogenic density (2,500 cells cm$^{-2}$), and grown in non-adherent culture conditions, as described[7]. In detail, cells were grown for 7–10 days in DMEM:F12 (1:2) supplemented with B27 (Invitrogen Corporation, Carlsbad, CA, USA), 20 ng ml$^{-1}$ EGF (PROSPEC, East Brunswick, NJ, USA), 20 ng ml$^{-1}$ bFGF (BD Biosciences, San Jose, CA, USA), 4 μg ml$^{-1}$ heparin (StemCell Technologies Inc.), 0.5 μg ml$^{-1}$ hydrocortisone (Sigma) and 5 μg ml$^{-1}$ Insulin (Sigma) in low attachment 24- or 96-well plates (Corning) in a humified incubator at 37 °C, 5% CO$_2$. Primary mammospheres (≥200 μm) were obtained, collected, counted and again enzymatically disaggregated as above to re-plate cells at clonogenic densities to obtain secondary mammospheres. Percentages of mammosphere forming efficiencies (%MFE) were calculated as number of mammospheres (≥200 μm) divided by the plated cell number and multiplied by a hundred. Mammospheres were counted with a ×20 objective on an Olympus CK30 microscope (Olympus Italia Srl, Milan, Italy).

**Flow cytometric analyses.** FACS analyses of CD44/CD24/EpCAM surface markers were performed as described[7,32]. Non-confluent cell cultures were trypsinized into single-cell suspension, counted, washed with phosphate-buffered saline (PBS), and blocked in PBS + BSA 1% + EDTA 2 mM. Staining with antibodies was performed for 30 min at 4 °C and cells were analysed by BD FACSCelesta Cell Analyzer (BD Biosciences). For CellROX analysis, cells were treated with KPT-6566, NAC or DMSO. After 48 h, CellROX Green Reagent (Thermo Fisher Scientific) was added at a final concentration of 5 μM for 30 min. Cells were then washed, trypsinized and analysed by FACS. Mean fluorescence intensity was derived from each sample and the baseline fluorescence was

calculated from the sample that was not incubated with CellROX reagent. For Annexin V/Propidium Iodide analysis, Alexa Fluor 488 Annexin V/Dead Cell Apoptosis Kit (Invitrogen) was used following the manufacturer's protocols. Analyses were performed on a FACSCalibur cell sorter (Becton Dickinson) and data were analysed with FlowJo software for Mac (FlowJo, LLC 2013-2016).

**Microarray hybridization and low level analysis.** For gene expression profiling in MDA-MB-231 cell lines, we used the Illumina HumanHT-12-v4-BeadChip (Illumina). Total RNA isolated from the cell lines with control siRNA or *PIN1* siRNA or treated with DMSO or KPT-6566 for 48 h were reverse transcribed and amplified, according to standard protocols and *in vitro* transcription was then carried out to generate cRNA. cRNA was hybridized onto each array (three biological replicates for the KPT-6566 condition and two for *PIN1* siRNA condition) and then labelled with Cy3-streptavidin (Amersham Biosciences). The array was then scanned using a BeadStation 500 system (Illumina). The probe intensities were calculated and normalized using GenomeStudio Data Analysis Software's Gene Expression Module (GSGX) Version 1.9 (Illumina). Further data processing was performed in the R computing environment version 3.2 (http://www.r-project.org/), with BioConductor packages (http://www.bioconductor.org/).

**Functional annotation and pathway enrichment analysis.** Full transcriptomic expression data sets have been imported to IPA software (Qiagen, www.ingenuity.com). *P* value and log fold-change cutoffs were applied in IPA as reported in the text and figures. Pathway analysis module of IPA was further used to associate analysed gene lists with molecular pathways (Fig. 5c). An independent, parallel method for analysing the signatures was the pathway-related gene ontology term enrichment analysis, using DAVID/EASE web tool with default parameters and procedures (Fig. 5b and Supplementary Table 9).

**Animal studies.** Experiments were performed with 6-week-old female, Hsd:Athymic Nude-*Foxn1$^{nu}$* mice obtained from Envigo (Udine, Italy). Mice were maintained under specific-pathogen-free conditions with constant temperature and humidity, according to institutional guidelines. Animal experimentation was conducted in conformance with the following laws, regulations and policies governing the care and use of laboratory animals: Italian Governing Law (D.lgs 26/2014; Authorization no. 19/2008-A issued March 6, 2008 by the Ministry of Health). Mario Negri Institutional Regulations and Policies providing internal authorizations for people conducting animal experiments (Quality Management System Certificate–UNI EN ISO 9001:2008—Reg. N° 6121); the NIH Guide for the Care and Use of Laboratory Animals (2011 edition) and EU directives and guidelines (EEC Council Directive 2010/63/UE), and in line with guidelines for the welfare and use of animals in cancer research[58]. Animal experimental protocols were reviewed and approved by the IRFMN Animal Care and Use Committee (IACUC), which include 'ad hoc' members for ethical issues. Animals were housed in the Institute's Animal Care Facilities, which meet international standards and were checked regularly by a certified veterinarian responsible for health monitoring, animal welfare supervision, revision of experimental protocols and procedures. For toxicity experiments mice (three for each group) were treated (i) intravenous (i.v.) with vehicle (8% DMSO, 8% TWEEN80, 84% PBS) or 60 and 90 mg kg$^{-1}$ of KPT-6566 every 2 days for three times; (ii) i.p. with vehicle or 30 and 45 mg kg$^{-1}$ of KPT-6566 every 2 days for 2 weeks. For lung colonization experiments mice were injected i.v. with 1 million of MDA-MB-231Luc[6] cells. After 1 day, mice were randomized in two experimental groups (at least seven mice for each group) and treated i.p. with vehicle (4% DMSO, 4% TWEEN80, 92% PBS) or KPT-6566 5 mg kg$^{-1}$, every day until killed (26 days of treatments). The investigators were not blinded to group allocation during experiments and outcome. During the study mice were monitored for parameters including appearance and body weight. Mice were killed when they became moribund. Lungs and other organs were collected, weighted and formalin fixed for histological analysis.

**Histology and metastatic area determination.** For lung metastases histology, 3 μm sections were cut from the lung lobes and stained with haematoxylin and eosin. For computer-aided assessment of lung tissue, the area occupied by metastatic foci, identifiable in sections of lung lobes, were calculated using a Leica Aperio Digital Pathology Slide Scanner and the software Leica Aperio Image Scope, summed and total value was finally compared to the whole area of the lung lobe.

**Statistical analyses.** The sample size was chosen to include at least three biological replicates. No statistical method was used to predetermine the sample size for animal studies. Experiments for which we showed representative images were performed successfully at least three independent times. No samples or animals were excluded from the analysis. Standard laboratory practice randomization procedure was used for cell line groups and animals of the same age and sex. The investigators were not blinded to allocation during experiments and outcome.

Statistical tests were appropriately chosen for each experiment. For differentially expressed genes from microarray experiments statistical analysis was performed

with limma and P values were adjusted for multiple testing using Benjamini and Hochsberg's method to control the false discovery rate. In the lung colonization assay P value was determined using two-tailed Mann–Whitney tests and statistical significance was set at $P < 0.05$. For all other experiments P values were determined using two-tailed Student's t-test and statistical significance was set at $P < 0.05$. An estimation of the variation within each group of data was indicated by s.d., std. error, s.e.m. or c.i. The variance was similar between groups that we compared.

**Data availability.** The authors declare that the main data supporting the findings of this study are available within the article and its Supplementary Information file. Full transcriptomic expression data sets are available at GEO database (record GSE84909). Extra data are available from the corresponding author on request.

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

## Acknowledgements

We thank A. Testa for reading and editing the manuscript. We acknowledge
G. Pastore for technical support; O. Gileadi (SGC Oxford) for pNIC28-Bsa4 vector;
B. Belletti (Department of Translational Research, CRO National Cancer Institute,
Aviano, Italy) for providing HMEC; B. Shilton (University of Western Ontario,
London Ontario, Canada) for discussion. We acknowledge the Centre for Personalized
Diagnostics and Arizona State University plasmid repository (DNASU) for GST-FKBP4
and GST-PPIA expression plasmids. Work in our lab is supported by the Italian
Association for Cancer Research (AIRC) Special Program Molecular Clinical
Oncology '5 per mille' (Grant no. 10016), AIRC IG (Grant no. 17659), the Cariplo
Foundation (Grant no. 2014-0812), the Italian Health Ministry (RF-2011-02346976),
Beneficentia-Stiftung, and Fondazione CRTrieste to G.D.S.; funds from the Italian
University and Research Ministry (PRIN-2015-8KZKE3) to G.D.S., M.L.L. and A.C.P.;
the European Fund for Regional Development-Cross-Border Cooperation Programme
Italy-Slovenia 2007–2013, (Project PROTEO, Code no. CB166) to P.S., B.G. and G.D.S.;
AIRC IG (Grant no. 201314414) to F.B.; E.C. is recipient of a fellowship from the
European Social Fund, POR 2007/13 Friuli Venezia Giulia, OB.2, AS. 5, AZ.85,
Mobilità Transnazionale, Regional Code FP1340305003; Y.C. is recipient of an
AIRC fellowship.

## Author contributions

E.C. and A.R. designed and performed experiments and wrote the manuscript. A.Z. and
A.C. performed experiments, S.P. and Y.C. the bioinformatics analyses; O.K. and G.G.
performed the virtual screening and initial cytotoxicity tests; E.B. and S.S. supervised the
drug screening and the hit discovery process; B.V. and U.C. provided all MS analyses;
A.C.P. and M.L.L. were in charge for compound synthesis; B.G. and P.S. performed
cloning and recombinant protein productions for *in vitro* and MS analyses; G.R. and P.C.
performed the molecular modelling; F.B. was in charge for FACS analyses; E.B., M.D.I.,
E.C. and A.R. were responsible for treatments and analysis of xenografted mice. G.D.S.
wrote the manuscript and supervised the overall project.

## Additional information

naturecommunications

**Competing interests:** Erkan Baloglu and Sharon Shacham are employees and
shareholders of Karyopharm Therapeutics. The remaining authors declare no competing
financial interests.

**Reprints and permission** information is available online at http://npg.nature.com/
reprintsandpermissions/

targets cancer cells by a dual mechanism of action. *Nat. Commun.* **8,** 15772
doi: 10.1038/ncomms15772 (2017).

**Publisher's note:** Springer Nature remains neutral with regard to jurisdictional claims in
published maps and institutional affiliations.

