## [Peer Review File · Nature Communications]

Reviewers' comments:

Reviewer #1 (Remarks to the Author):

This paper claims to identify a novel covalent inhibitor of Pin1 that upon reaction with Pin1 Cys113 releases a sulphonamide-quinone compound that elicits reactive oxygen generation and causes DNA damage. The compound therefore both inhibits Pin1 (indeed it induces degradation) and releases toxins specifically in cells expression reasonably high levels of Pin1 (perhaps with some specificity for cells where Pin1 inhibitory phosphorylations are not in play). The authors have, in general, provided a set of data that provides strong support for their conclusions. I also think such a compound is of potential interest to the wider anti-cancer drug community. As Pin1 drug discovery is somewhat obscure nowadays, the field will benefit for a reasonably high impact publication in Nature Comms.

Personally I find the proliferation of quantified western blots in the literature as rather futile, as is the provision of p values on bar charts showing obvious differences. The methods section (around line 812) described the statistical tests performed in sufficient detail for me and they are as far as I can tell appropriate.

As for the ability of others to reproduce the work, I don't see many missing pieces of information. However, I couldn't find the concentration of KPT compounds that was used in the S375 vs 3T3 specific cytotox screen.

The only suggestions I have for this paper are to restate some of the facts from the literature with more precision and to qualify some of the statements made. these minor points relate to:

Line 56. It would be helpful to explain that Pin1 is a bi-directional catalyst. It allows the conformation changes driven by charge changes introduced by phosphorylation/dephosphorylation to actually occur at a sensible rate: the S-P or T-P bond only slowly rotates.

Line 67. I don't agree with the statement that Pin1 is an amplifier. Modulator would be OK, especially if one refers back to the changes I'd like to be made around line 56.

Line 69. Pin1 should enable - not promote

Line 72: Pin1 is not essential for tumour growth. pin1 KO mice exist and they are not completely tumour free. In a similar vein, forced expression of Myc is able to transform immortalised Pin1-deficient mammary cells. (Pin1 does seem to be a lot more important for ErbB2 mediated tumourigenesis).

Line 7. The authors should check ref 9 which from the title implies sorafenib is inhibiting Pin1. Does that paper really show that???

Line 80. Pin1 is actually very challenging for the design of cell permeable small molecule inhibitors (small shallow pocket with lysines to bind the phosphorylated Ser/thr residue) and the best assays are very difficult to run at scale.

Line 89. The term selectively should be qualified (eg amongst PPIases)

Line 149. Does docking data ever indicate?? I would prefer suggest.

Line 156. The sulfanyl-acetate moiety has merely been modelled deep inside the catalytic pocket

Line 167. If the authors want to say their compound has a high kinetic, I think this needs some context provision re other irreversible inhibitor/enzyme combinations.

Line 183. The cells treated with 5 uM compound don't look almost arrested to me.

Line 194. The compound had only a modest effect on empty vector expressing cells

Line 200. Please qualify how ref 10 shows that Pin1 is mainly inactive in ref 10

Line 258. Significant would be a fairer description than robust (a proper robust effect is shown in Fig 3g).

Line 262. Does 2 fold represent a STRONG decrease in M2FE?

Line 272. This result implies that the role of pin1 in M2FE in MDA-MB-231 cells is not clearcut.

Line 312. Conspicuous similarity? With only 34% of changes (and just the down ones) in common. This statement seems a little strong.

Line 373. The reference to the literature that Pin1 is important to the DDR seems at odds with the results in this paper.

Reviewer #2 (Remarks to the Author):

This is a well written manuscript presenting extensive characterisation of a novel lead compound targeting PIN1. It will be of interest to the general readership of this Journal, due to the in silico screening technology described, in addition to the disease and gene specific effects that are detailed. This is a major advance.

Major point.

1. The section on mammary stem cells and self renewal over interprets the sphere forming assay. Mammary stem cells are not exclusively defined by the CD24/CD44 markers used, nor is sphere formation a definitive test of stemness or self renewal capacity. Limiting dilution transplants in mice are the current method of assessing stem cell activity. The conclusions here should be restricted to an influence on sphere formation and cell surface makers as the data does not support the current conclusions. Only changes to the text are required to do this.

Minor point.

Checking of the grammar is required, numerous small errors are made.

Reviewer #3 (Remarks to the Author):

Campaner and collaborators describe the identification and characterisation of a new covalent PIN1 inhibitor that targets cancer cells by a dual mechanism of action. As the authors indicate in the manuscript, there is a selection of PIN1 inhibitors that have been shown to affect tumorigenesis in different systems. However, the authors claim that this novel one is more specific, targeting cancer cells, while sparing normal cells. In addition, they show a variety of assays to demonstrate that binding of the inhibitor results in DNA damage and cell death in vitro and, furthermore, impairs growth of lung metastasis in vivo.

This is an interesting study, although some of the conclusions are not sufficiently supported by the data presented. The manuscript would benefit from additional experiments, as indicated below:

- The assays with the potential inhibitors used A375 cells and normal 3T3 as controls. The effect (or lack of) of the selected inhibitor should be shown in both cell types (presently as "data not shown") to confirm lack of toxicity in 3T3 cells.
- The authors concluded that their results "clearly indicated that KPT-6566 selectively impairs Pin1 PPIase activity", although the control experiments were done with just 2 other kinases and, in fact, later on they showed (Fig 5) that there is a transcriptional program for the inhibitor that is independent of the presence of PIN1. This should be discussed.
- Following this point, it is a little puzzling to see that despite their statement that the inhibitor is specific for cancer cells, most of figure 2 is dedicated to assess the effects of the inhibitor on MEFs, which are non-tumorigenic cells. It appears that the effects of the inhibitor may be linked to the levels of PIN1 expression and, therefore, it would be necessary to show a comparison of the effects of the inhibitor on various breast cancer cell lines in parallel and also in HBECs, not just limited to MCF10A. In addition, figure 2f must show a western blot to demonstrate the possible correlation between PIN1 levels and efficiency of the inhibitor in the different lines used.
- There is a clear increase in the levels of cyclin D1 in Pin1 KO MEFs (Fig 2b), please explain. The data with phospho-Rb is not convincing, taking into account the differences in total Rb there is a clear effect of the inhibitor in the KO cells. Please address this issue.
- All experiments presented with siRNA and shRNA for PIN1 show results with only one sequence. Functional effects of the inhibitor should be shown with more than one siRNA and shRNA.
- Throughout the manuscript the authors have used 5 microM (or 2.5 microM in some cases, with only modest effects), however, the colony formation assays used 1 microM and the effects are more obvious than in the proliferation or cell death assays. What happens in this type of assay if 5 microM is used?
- Given the above point, photographs of the primary and secondary mammospheres must be shown. The size/parameter used to define a mammosphere must be stated in the materials and methods. PIN1 overexpression greatly enhanced cells with capacity to form mammospheres, but that assay, by itself, does not represent stem cell number (Fig 3g). Furthermore, following their statement of MCF10A cells having a very low M2FE and representing stem cell number, they measured CD44CD24 expression and found 15.7% in MCF10A, which would be a very high % of stem cells. To address the potential effect of PIN1 levels in the CD44+CD24-population of stem cells they must use ESA as well for MCF10A cells (see Fillmore and Kuperwasser. 2008) and add error bars. Finally, if the authors aim to claim effects of the inhibitor on the stem cell population, they must also analyse potential changes in the cancer cell lines. Presently this conclusion has not been demonstrated.
- Lane 269 claims that the PIN1-dependent gain of function was abolished by the inhibitors in a dose-dependent manner, but only one concentration was used in the figures (3f, g).
- The effects of the inhibitor on PIN1 mRNA or PIN1 promoter are dismissed as "only slightly reduced". The experiments with PIN1 promoter should be shown and the effects on PIN1 mRNA are statistically significant. The significance of this regulation should be discussed.
- PC3 cells appear to have considerable levels of active PIN1 (e.g. see Fig sup 3), however, they do not show a significant H2AX phosphorylation. Please address this issue. In addition, H2AX phosphorylation should also be examined in MEFs, which also express active PIN1, as demonstrated by the authors, to determine whether these effects are limited to cancer cells.
- The effects of the inhibitor on ROS levels should also be determined in normal cells, including HBECs and MEFs, as well as in PC3 cells.

Rebuttal on revision of manuscript NCOMMS-16-20289:

As requested by the reviewers, we have changed and improved several parts of the text and performed additional experiments to meet all the concerns. Modified text appears in red in the revised version of the manuscript. Details are listed below in point-by-point answers (text in italics).

Reviewer #1 (Remarks to the Author):

This paper claims to identify a novel covalent inhibitor of Pin1 that upon reaction with Pin1 Cys113 releases a sulphonamide-quinone compound that elicits reactive oxygen generation and causes DNA damage. The compound therefore both inhibits Pin1 (indeed it induces degradation) and releases toxins specifically in cells expression reasonably high levels of Pin1 (perhaps with some specificity for cells where Pin1 inhibitory phosphorylations are not in play). The authors have, in general, provided a set of data that provides strong support for their conclusions. I also think such a compound is of potential interest to the wider anti-cancer drug community. As Pin1 drug discovery is somewhat obscure nowadays, the field will benefit for a reasonably high impact publication in Nature Comms.

Personally I find the proliferation of quantified western blots in the literature as rather futile, as is the provision of p values on bar charts showing obvious differences. The methods section (around line 812) described the statistical tests performed in sufficient detail for me and they are as far as I can tell appropriate.

Answer: According to this suggestion, we have removed the p-values where the differences were obvious. We have also removed the majority of the band quantifications.

As for the ability of others to reproduce the work, I don't see many missing pieces of information. However, I couldn't find the concentration of KPT compounds that was used in the S375 vs 3T3 specific cytotox screen.

Answer: We have now added this information in the Methods section. In addition, the IC50 values of the compounds used in the cytotoxicity experiments in A375 and 3T3 cells are now shown in the Supplementary Table 1.

The only suggestions I have for this paper are to restate some of the facts from the literature with more precision and to qualify some of the statements made. these minor points relate to:

Line 56. It would be helpful to explain that Pin1 is a bi-directional catalyst. It allows the conformation changes driven by charge changes introduced by phosphorylation/dephosphorylation to actually occur at a sensible rate: the S-P or T-P bond only slowly rotates.

Answer: To better explain how PIN1 catalyzes the cis/trans isomerization reaction we have revised the text in the introduction, as follows: "In proteins, S/T-P motifs can adopt either a cis or a trans conformation. Spontaneous conversion between isomers occurs at a very slow rate and is further slowed down by phosphorylation of these motifs. However, phospho-S/T-P sites can be recognized by the peptidyl-prolyl cis/trans isomerase (PPIase) PIN1, which catalyzes cis-trans or trans-cis conformational changes around the S-P or T-P bond. Among PPIases, PIN1 is the only enzyme able to efficiently bind proteins containing phosphorylated S/T-P sites."

Line 67. I don't agree with the statement that Pin1 is an amplifier. Modulator would be OK, especially if one refers back to the changes I'd like to be made around line 56.

Answer: We agree and replaced "amplifier" with "modulator".

Line 69. Pin1 should enable - not promote

Answer: We agree and replaced "promote" with "enable".

Line 72: Pin1 is not essential for tumour growth. pin1 KO mice exist and they are not completely tumour free. In a similar vein, forced expression of Myc is able to transform immortalised Pin1-deficient mammary cells. (Pin1 does seem to be a lot more important for ErbB2 mediated tumorigenesis).

Answer: We agree with the reviewer's comments. We now modified the text as follows: "Genetic ablation of PIN1 reduces tumor growth and metastasis in several oncogene-induced mouse models of tumorigenesis, indicating the requirement of PIN1 for both tumor development and progression."

Line 7. The authors should check ref 9 which from the title implies sorafenib is inhibiting Pin1. Does that paper really show that???

Answer: *This reference was at the wrong place and now has been moved to the correct position. The demonstration that PIN1 inhibition overcomes chemoresistance was shown by us in Rustighi, et al. EMBO Mol Med. 2014, 6:99-109. We now added the reference in the new version of the manuscript.*

Line 80: pin1 is actually very challenging for the design of cell permeable small molecule inhibitors (small shallow pocket with lysines to bind the phosphorylated Ser/thr residue) and the best assays are very difficult to run at scale.

Answer: *Following this point we modified the text as follows: "However, its small and shallow enzymatic pocket, as well as the requirement of a molecule with a negatively charged moiety for interfacing with its catalytic center have been challenging the design of PIN1 inhibitors."*

Due to space constraints, we did not address the difficulties of the PIN1 enzymatic assays, but we referred to the very exhaustive review of Moore & Potter, BMCL 2013, 23: 4283-91.

Line 89. The term selectively should be qualified (eg. amongst PPIases)

Answer: *The selectivity criterion is specified in the results and in the discussion sections.*

Line 149. Does docking data ever indicate?? I would prefer suggest.

Answer: *We changed the text according to this comment.*

Line 156. The sulfanyl-acetate moiety has merely been modelled deep inside the catalytic pocket

Answer: *As correctly observed by the reviewer, the cavity does not perfectly accommodate the sulfanyl-acetate moiety, possibly because the ligand is very bulky (Fig. 1e). However, it may further enter the cavity when the covalent reaction with PIN1 has occurred. Therefore, we removed the whole sentence from the text and added the docking scores and related result we obtained (Supplementary Table 3), in which, the binding pose we suggested (now added as "Pose 1" in Supplementary Table 3) is the best one according to Glide Docking Score.*

Line 167. If the authors want to say their compound has a high kinact, I think this needs some context provision re other irreversible inhibitor/enzyme combinations.

Answer: We have now added that K_{inact} of KPT-6566 is higher than those of other covalent PPIase inhibitors, such as Juglone (Hennig et al., *Biochemistry* 1998, 37: 5953- 5960).

Line 183. The cells treated with 5 μ M compound don't look almost arrested to me.

Answer: We have now modified the sentence and wrote that: "In WT MEFs KPT-6566 had a negative, dose-dependent effect on proliferation."

Line 194. The compound had only a modest effect on empty vector expressing cells.

Answer: This comment refers to Fig. 2c, where Pin1 knockout MEFs were employed to assess whether the effect of KPT-6566 on proliferation was dependent on PIN1. In fact, when these cells ectopically express PIN1 (right panel), they become sensitive again to KPT-6566 treatment, whereas empty-vector cells do not (left panel). The slight difference between untreated and treated Pin1 knockout MEFs with empty vector is not statistically significant, therefore we now have modified the text as follows: "In empty-vector expressing Pin1 KO MEFs, instead, KPT-6566 had only a slight and statistically not significant effect on proliferation and no impact on Cyclin D1."

Line 200. Please qualify how ref 10 shows that Pin1 is mainly inactive in ref 10

Answer: We have added an explanation, as follows: "We next tested the effects of increasing doses of KPT-6566 on an isogenic cell model constituted of i) non-transformed MCF10A mammary epithelial cells, which express lower levels of PIN1 than cancer cell lines, and in which the enzyme is mainly inactivated due to S71 phosphorylation, and ii) H-RasV12-transformed MCF10AT1 cells which, instead, express high levels of active PIN1."

Line 258. Significant would be a fairer description than robust (a proper robust effect is shown in Fig 3g).

Answer: We agree and we wrote "significant" instead of "robust".

Line 262. Does 2 fold represent a STRONG decrease in M2FE?

Answer: We wrote “twofold” instead of “strong” which is more precise.

Line 272. This result implies that the role of pin1 in M2FE in MDA-MB-231 cells is not clearcut.

Answer: M2FE of KPT-6566 treated MDA-MB-231 cells showed a twofold reduction compared to M2FE of DMSO treated cells. This result is in accordance with previously published data (Rustighi et al., EMBO Mol Med. 2014, 6:99-109) showing that M2FE of MDA-MB-231 cells is affected by knockdown of PIN1 or by treatment with other PIN1 inhibitors.

Line 312. Conspicuous similarity? With only 34% of changes (and just the down ones) in common. This statement seems a little strong.

Answer: We now have removed “conspicuous”.

Line 373. The reference to the literature that Pin1 is important to the DDR seems at odds with the results in this paper.

Answer: We agree, and, as this sentence is confounding, we removed it.

Reviewer #2

This is a well written manuscript presenting extensive characterisation of a novel lead compound targeting PIN1. It will be of interest to the general readership of this Journal, due to the in silico screening technology described, in addition to the disease and gene specific effects that are detailed. This is a major advance.

Major point. 1. The section on mammary stem cells and self renewal over interprets the sphere forming assay. Mammary stem cells are not exclusively defined by the CD24/CD44 markers used, nor is sphere formation a definitive test of stemness or self renewal capacity. Limiting dilution transplants in mice are the current method of assessing stem cell activity. The conclusions here should be restricted to an influence on sphere formation and cell surface makers as the data does not support the current conclusions. Only changes to the text are required to do this.

Answer: We agree with the reviewer and therefore we modified the text describing the effect of KPT-6566 only on mammosphere formation efficiency and on expression of stem cell markers.

Minor point. Checking of the grammar is required, numerous small errors are made.

Answer: The text has been revised.

Reviewer #3

Campaner and collaborators describe the identification and characterisation of a new covalent PIN1 inhibitor that targets cancer cells by a dual mechanism of action. As the authors indicate in the manuscript, there is a selection of PIN1 inhibitors that have been shown to affect tumorigenesis in different systems. However, the authors claim that this novel one is more specific, targeting cancer cells, while sparing normal cells. In addition, they show a variety of assays to demonstrate that binding of the inhibitor results in DNA damage and cell death in vitro and, furthermore, impairs growth of lung metastasis in vivo.

This is an interesting study, although some of the conclusions are not sufficiently supported by the data presented. The manuscript would benefit from additional experiments, as indicated below:

The assays with the potential inhibitors used A375 cells and normal 3T3 as controls. The effect (or lack of) of the selected inhibitor should be shown in both cell types (presently as “data not shown”) to confirm lack of toxicity in 3T3 cells.

Answer: We have now added the IC50 values of the compounds used in the cytotoxicity experiments in A375 and 3T3 cells in the Supplementary Table 1.

The authors concluded that their results “clearly indicated that KPT-6566 selectively impairs Pin1 PPIase activity”, although the control experiments were done with just 2 other kinases and, in fact, later on they showed (Fig 5) that there is a transcriptional program for the inhibitor that is independent of the presence of PIN1. This should be discussed.

Answer: We thank the reviewer for this comment and we would like to better clarify this point. We evaluated the selectivity of KPT-6566 for PIN1 by testing in vitro the inhibitory effect of the drug on the enzymatic activity of PIN1 and of PPIA and FKBP4. The latter enzymes are representatives of the other two existing families of Peptidyl Prolil Isomerases (PPIase) and, like PIN1, they contain Cysteines in their catalytic domains. The results of this experiment let us conclude that KPT-6566 behaves as a selective inhibitor of PIN1 and not of other PPIases (Figs. 1g,h). A similar approach has already been adopted to assess the selectivity of other PIN1 inhibitors, such as Juglone (Hennig et al., Biochemistry 1998, 37: 5953- 5960) and Dipentamethylene thiuram monosulfide (Tatara et al., BBRC

2009, 384:394-8). In the new version of the manuscript we have improved the description of this result.

Regarding the reviewer's comment: "that there is a transcriptional program for the inhibitor that is independent of the presence of PIN1", we would like to clarify that we have compared the transcriptional profile of cells treated with KPT-6566 with that of cells silenced for PIN1. This analysis revealed that the transcriptional profiles related to these two different conditions significantly overlap, likely due to the direct inhibitory effect of KPT-6566 on PIN1. In addition, this analysis unveiled also non-overlapping transcriptional profiles, suggesting that KPT-6566 might elicit other effects besides PIN1 inhibition. Functional analysis of the differentially regulated genes between the two conditions revealed KPT-6566 specific effects, related in particular to oxidative stress response and DNA damage. Although we did not perform experiments to assess the requirement of PIN1 for these transcriptomic alterations, we were able to show (Fig. 6) that at least the DNA damaging activity of KPT-6566 is dependent on the presence of PIN1.

Following this point, it is a little puzzling to see that despite their statement that the inhibitor is specific for cancer cells, most of figure 2 is dedicated to assess the effects of the inhibitor on MEFs, which are non-tumorigenic cells.

Answer: The experiment performed in Pin1 WT and KO MEFs (Fig. 2a) is a well-established assay to assess specificity of PIN1 inhibitors (Uchida et al., Chem Biol. 2003, 10:15-24; Wei et al., Nat Med. 2015, 21:457-66). MEFs are known to depend on PIN1 function for their proliferation (Tong et al., Cell Death & Disease 2015, 6:e1640) and thus we used them to demonstrate that, by inhibiting PIN1, KPT-6566 specifically affects proliferation of Pin1 WT cells. The specificity for cancer cells we described refers to the induction of cell death specifically in cancer cells treated with KPT-6566, while in MEFs we observed only inhibition of proliferation upon treatment.

It appears that the effects of the inhibitor may be linked to the levels of PIN1 expression and, therefore, it would be necessary to show a comparison of the effects of the inhibitor on various breast cancer cell lines in parallel and also in HBECs, not just limited to MCF10A. In addition, figure 2f must show a western blot to demonstrate the possible correlation between PIN1 levels and efficiency of the inhibitor in the different lines used.

Answer: We thank the reviewer for the comment and we have performed the suggested experiments which are now in Figs. 2f (Analysis of PIN1 levels of the cell lines of Fig. 2g) and 2g (Viability in human epithelial cell lines) of the new version of the manuscript. As shown in Fig. 2g, the IC50 of KPT-6566 is higher in normal cells than in cancer cells and these latter display higher levels of PIN1.

There is a clear increase in the levels of cyclin D1 in Pin1 KO MEFs (Fig 2b), please explain. The data with phospho-Rb is not convincing, taking into account the differences in total Rb there is a clear effect of the inhibitor in the KO cells. Please address this issue.

Answer: The Western blot analysis has been performed again with normalized actin levels, and we have replaced the old panels in Fig. 2b. These new data show that in untreated cells, Cyclin D1 levels are comparable in WT and Pin1 KO MEFs, whereas upon KPT-6566 treatment protein levels of Cyclin D1 and phospho-pRb are decreased only in WT cells and unaffected in Pin1 KO MEFs.

All experiments presented with siRNA and shRNA for PIN1 show results with only one sequence. Functional effects of the inhibitor should be shown with more than one siRNA and shRNA.

Answer: We agree with the reviewer's concern and we have now performed our main functional experiments with another siRNA sequence and repeated i) experiments of colony formation in PC-3 and MDA-MB-231 cells and ii) analysis of H2A.X phosphorylation. These experiments provided nearly identical results to those obtained with PIN1 siRNA #1 and are now included in the new Supplementary Figures 3b-d, and 3f and Figure 6c-e.

Throughout the manuscript the authors have used 5 microM (or 2.5 microM in some cases, with only modest effects), however, the colony formation assays used 1 microM and the effects are more obvious than in the proliferation or cell death assays. What happens in this type of assay if 5 microM is used?

Answer: Regarding this point, we performed a KPT-6566 dose-response curve experiment to assess which was the concentration leading to a 50% reduction of the colony forming ability. As shown in the dose-response curves (Supplementary Fig. 3a) the 5 μ M concentration of KPT-6566 almost completely abolished colony formation. We used 1 μ M KPT-6566 in colony formation experiments since this was the concentration that reduced colony formation to 50% (Figure 3c, and Suppl. Figs. 3c,e,f).

Given the above point, photographs of the primary and secondary mammospheres must be shown. The size/parameter used to define a mammosphere must be stated in the materials and methods. PIN1

overexpression greatly enhanced cells with capacity to form mammospheres, but that assay, by itself, does not represent stem cell number (Fig 3g).

Furthermore, following their statement of MCF10A cells having a very low M2FE and representing stem cell number, they measured CD44CD24 expression and found 15.7% in MCF10A, which would be a very high % of stem cells. To address the potential effect of PIN1 levels in the CD44+CD24- population of stem cells they must use ESA as well for MCF10A cells (see Fillmore and Kuperwasser. 2008) and add error bars. Finally, if the authors aim to claim effects of the inhibitor on the stem cell population, they must also analyse potential changes in the cancer cell lines. Presently this conclusion has not been demonstrated.

Answer: As requested by the reviewer, we have now added representative photographs of secondary mammospheres (Supplementary Figures 3h,k,m of the new version of the manuscript) as well as reference of their size in the Methods section. Moreover, according to the reviewer's comment, we have modified the text and have described the effect of KPT-6566 on sphere formation and on expression of stem cell markers.

Regarding the second point, we repeated the FACS analysis of CD24/CD44 cell surface markers in order to provide new data with standard deviation and statistical analysis. The data that we obtained (Supplementary Table 6 of the new version of the manuscript) are consistent with other published results (Fillmore & Kuperwasser, Breast Cancer Research 2008, 10:R25; Rustighi et al., EMBO Mol Med. 2014, 6:99-109). As expected, the overexpression of PIN1 in MCF10A cells causes an enrichment of the CD44⁺/CD24⁻ population, indicating that PIN1 is promoting an epithelial-to-mesenchymal transition (EMT) program that confers stem cell traits (as demonstrated in Rustighi et al., EMBO Mol Med. 2014, 6:99-109).

Regarding the use of the surface antigen ESA (EpCAM), known to be a stem cell marker of breast cancer cells (Al-Hajj et al., PNAS 2003, 100:3983-88; Fillmore and Kuperwasser et al., Breast Cancer Research 2008, 10:R25), recent studies demonstrated lack of correlation between the expression of ESA and stemness features in non-transformed MCF10A cells (Liu et al., Stem Cell Reports 2014, 2:78-91). For this reason we did not analyze ESA expression in MCF10A in these experiments.

Finally, in accordance to the reviewer's comment regarding the effect of KPT-6566 on the cancer stem cell population, we have now analyzed the expression levels of well-known stemness factors such as Slug, Sox9 and Oct4 (Guo et al., Cell 2012, 148:1015-28; Ben-Porath et al., Nat Genet. 2008, 40:499-507) in MDA-MB-231 breast cancer cells. This analysis shows a decrease of the protein levels of all these stemness factors upon treatment with KPT-6566 (Figure 3f of the revised version). In addition, we have performed FACS analyses of CD44/CD24/ESA in cancer cells (MDA-MB-231) treated with KPT-6566. The result showed that KPT-6566 treatment exerts a negative effect on the CD44⁺/CD24^{-low}/ESA⁺ population, supporting the idea that KPT-6566 is affecting cancer stem cell traits. These data are now presented in Supplementary Fig. 3i of the new version of the manuscript.

Lane 269 claims that the PIN1-dependent gain of function was abolished by the inhibitors in a dose-dependent manner, but only one concentration was used in the figures (3f, g).

Answer: The missing experimental point has been added in the figure (Fig. 3h of the revised version).

The effects of the inhibitor on PIN1 mRNA or PIN1 promoter are dismissed as “only slightly reduced”. The experiments with PIN1 promoter should be shown and the effects on PIN1 mRNA are statistically significant. The significance of this regulation should be discussed.

Answer: We have now added the experimental data on PIN1 promoter activity upon KPT-6566 treatment (Supplementary Fig. 4a). As can be observed in Suppl. Figs. 4 a,b both PIN1 promoter activity and mRNA levels are slightly downregulated by KPT-6566. It is important to note that PIN1 promotes the activity of three transcriptional factors (Notch1, Notch4, and E2F) that have been shown to bind to PIN1 promoter and increase its transcription (Ryo et al., Mol Cell Biol. 2002, 22(15):5281-95; Rustighi et al., Nat Cell Biol. 2009, 11:133-42; Rustighi et al., EMBO Mol Med. 2014, 6:99-109). Therefore it can be speculated that PIN1 inhibition by KPT-6566 impinges on this feed-forward loop producing the observed decrease in PIN1 promoter activity and transcription.

PC3 cells appear to have considerable levels of active PIN1 (e.g. see Fig sup 3), however, they do not show a significant H2AX phosphorylation. Please address this issue.

Answer: The different levels of H2A.X phosphorylation observed in the tested cell lines might reflect their different kinetics of DNA damage foci formation and repair (Macphail et al., Int J Radiat Biol 2003, 79:351-58; Banàth et al., Cancer Res. 2004, 64:7144-9.). This may be also inferred from H2A.X phosphorylation after Bleomycin treatment, a radiomimetic and direct inducer of DNA damage that did not elicit same levels of H2A.X phosphorylation in different cell lines.

In addition, H2A.X phosphorylation should also be examined in MEFs, which also express active PIN1, as demonstrated by the authors, to determine whether these effects are limited to cancer cells.

Answer: As requested by the reviewer, we have performed H2A.X phosphorylation analysis on MEFs treated with KPT-6566. We observed an increase in H2A.X phosphorylation in MEFs, likely due to the ability of KPT-6566 to increase ROS production, and thus to induce DNA damage. Mouse fibroblasts are known to have increased sensitivity to oxidative damage in culture (Parrinello et al., Nat Cell Biol. 2003, 5: 741–747), which could explain the observed sensitivity to KPT-6566 treatment. However, under these conditions we did not observe cell death, but only a reduction of

proliferation (Fig. 2a). We did not to include the H2A.X phosphorylation analysis in MEFs in the manuscript, and we provide the result here below for the reviewer's evaluation.

Immunoblotting of the indicated proteins from mouse embryo fibroblasts treated with 5 μ M KPT-6566, 10 μ M Bleomycin or DMSO (-) for 48hrs.

The effects of the inhibitor on ROS levels should also be determined in normal cells, including HBECs and MEFs, as well as in PC3 cells.

Answer: According to the reviewer's suggestion, additional experiments were performed to determine ROS levels in PC-3 and in non-transformed human breast epithelial cells (MCF10A), upon KPT-6566 treatment alone or in combination with NAC. Addition of KPT-6566 elicited an increase of ROS in both cancer cells and normal cells. However, in general the amount of ROS is five times lower in normal cells than in cancer cells. This result, together with the viability assays shown in Fig. 2g, suggests that cancer cells displaying higher levels of ROS than normal cells are more sensitive to ROS overload caused by KPT-6566 treatment. Results are shown in Suppl. Fig. 5c.

MEFs, instead, showed levels of ROS that were comparable to those observed in normal epithelial cells. We did not include this result in the manuscript, but we provide it here below for the reviewer's evaluation.

Histogram representing CellROX mean fluorescence intensity (MFI) of mouse embryo fibroblasts treated with 5 μ M KPT-6566 with or without 2.5 mM NAC or DMSO for 48hrs.

REVIEWERS' COMMENTS:

Reviewer #1 (Remarks to the Author):

The revised version of this paper answers the criticisms I had regarding the first version.

I would merely ask that the sentence in red in lines 73-75 be changed to end "requirement for PIN1 for the development and progression of some tumours4."

Spelling etc

Line 128 alanine should be alanine

Potentially useful suggestions.

Line 336. maybe it would be helpful to say the 6566-B compound acts "downstream" of Pin1

Line 376. Though the info is elsewhere in the paper. I think it would be helpful for the authors to say that the in vivo inhibitor treatment was daily for 27 days.

Reviewer #2 (Remarks to the Author):

Revisions meet my expectations, comments regarding cancer stem cells (CSC) are now functionally supported by sphere regrowth. I recommend publication.

Reviewer #3 (Remarks to the Author):

Campaner and collaborators have addressed most of the points suggested and the manuscript has considerably improved as a consequence.

NCOMMS-16-20289B

Point-by-point answers (text in italics) to the reviewers' comments:

Reviewer #1 (Remarks to the Author):

The revised version of this paper answers the criticisms I had regarding the first version.

I would merely ask that the sentence in red in lines 73-75 be changed to end "requirement for PIN1 for the development and progression of some tumours4."

Answer: we have modified the sentence according to this suggestion.

Spelling etc

Line 128 alanine should be alanine

Answer: we have introduced the correction.

Potentially useful suggestions.

Line 336. maybe it would be helpful to say the 6566-B compound acts "downstream" of Pin1

Answer: According to this suggestion we have introduced this concept in the discussion section as follows: "Most of all, however, KPT-6566 represents a one of a kind PIN1 inhibitor because it associates a highly specific PIN1 inhibitory activity with the release of a reactive quinone-mimicking byproduct that acts downstream of PIN1 and generates DNA damage and elicits cancer cell death."

Line 376. Though the info is elsewhere in the paper. I think it would be helpful for the authors to say that the in vivo inhibitor treatment was daily for 27 days.

Answer: We have now added this information in the text.

Reviewer #2 (Remarks to the Author):

Revisions meet my expectations, comments regarding cancer stem cells (CSC) are now functionally supported by sphere regrowth. I recommend publication.

Reviewer #3 (Remarks to the Author):

Campaner and collaborators have addressed most of the points suggested and the manuscript has considerably improved as a consequence.